# A monomeric mycobacteriophage immunity repressor utilizes two domains to recognize an asymmetric DNA sequence

Reliza J. McGinnis [1,6], Chad A. Brambley [2], Brandon Stamey[1], William C. Green[1], Kimberly N. Gragg[1], Erin R. Cafferty [1,7], Thomas C. Terwilliger [3], Michal Hammel [4], Thomas J. Hollis [5], Justin M. Miller [2], Maria D. Gainey [1✉] & Jamie R. Wallen [1✉]

Regulation of bacteriophage gene expression involves repressor proteins that bind and downregulate early lytic promoters. A large group of mycobacteriophages code for repressors that are unusual in also terminating transcription elongation at numerous binding sites (stoperators) distributed across the phage genome. Here we provide the X-ray crystal structure of a mycobacteriophage immunity repressor bound to DNA, which reveals the binding of a monomer to an asymmetric DNA sequence using two independent DNA binding domains. The structure is supported by small-angle X-ray scattering, DNA binding, molecular dynamics, and in vivo immunity assays. We propose a model for how dual DNA binding domains facilitate regulation of both transcription initiation and elongation, while enabling evolution of other superinfection immune specificities.

[1] Western Carolina University, Department of Chemistry and Physics, 111 Memorial Drive, Cullowhee, NC 28723, USA. [2] Middle Tennessee State University, Department of Chemistry, 1301 East Main Street, Murfreesboro, TN 37132, USA. [3] New Mexico Consortium, Los Alamos, NM 87544, USA. [4] Molecular Biophysics and Integrated Bioimaging, Lawrence Berkeley National Laboratory, Berkeley, CA 94720, USA. [5] Department of Biochemistry, Wake Forest University School of Medicine, Medical Center Boulevard, Winston-Salem, NC 27157, USA. [6] Present address: Department of Biomedical Engineering, University of Michigan, Ann Arbor, MI 48109, USA. [7] Present address: Department of Human Genetics, University of Utah School of Medicine, Salt Lake City, UT 84112, USA. ✉email: mdgainey@email.wcu.edu; jamiewallen@email.wcu.edu

Genomic characterization of bacteriophages lags that of their bacterial hosts. However, accumulating work from the Science Education Alliance Phage Hunters Advancing Genomics and Evolutionary Science (SEA-PHAGES) program is greatly expanding our knowledge of bacteriophage diversity and evolution[1–3]. As of this writing, the SEA-PHAGES program has annotated genomes of 3763 bacteriophages isolated from actinobacterial hosts. These viruses, which are classified into genetic clusters based both on nucleotide similarity and shared gene content[3], are a powerful resource to explore fundamental questions in bacteriophage biology and the development of antibacterial therapeutics.

Unlike most other viruses, the majority of bacteriophages in the environment have been found to be temperate, meaning that upon infection of a host, they can enter either the lytic or lysogenic cycles[4]. In the lysogenic cycle, the phage's genome is either integrated into the host chromosome or maintained extrachromosomally and replicated along with the host bacterial genome[5]. To maintain lysogeny and prevent entry into the lytic replication cycle, temperate bacteriophages encode a repressor protein that binds specific operator sites in the phage genome to repress the transcription of genes required for lytic growth[6]. Repressors can also provide the lysogen immunity to infection by other genetically similar bacteriophages whose genomes contain the same or similar operator sites[6]. The best characterized bacteriophage repressor is the cI protein from bacteriophage lambda[7]. cI binds to its operator sites as a dimer, but also forms homotetramers and homooctamers[7,8]. The N-terminus of cI (residues 1–92) contains a helix-turn-helix (HTH) motif that binds DNA, while the C-terminus (residues 132–236) is responsible for oligomerization[7,8]. cI dimers assemble such that each HTH domain engages half of a symmetric operator sequence at successive openings of the major groove on the same face of the DNA.

Actinobacteriophage repressor systems are distinct from the lambda-like phages[6], and to date repressors have been validated from clusters A, G, K, I, N, P, and Q[9]. Of these, only repressors from clusters A and G have been biochemically characterized[10–14], with the best characterized repressor system from mycobacteriophage L5, a cluster A bacteriophage isolated from *Mycobacterium smegmatis* mc[2]155[6,10,11]. Unlike lambda cI, the L5 repressor is predicted to bind as a monomer to more than twenty 13-mer asymmetric DNA-binding sites located throughout the genome. The repressor sites are found both in promoter regions (operators) as well as in short intergenic spaces (termed stoperators), with these sites proposed to halt transcription initiation and elongation, respectively[10]. The orientation of the asymmetric sequences shows a striking correlation with the direction of transcription. If the orientation of these sites is reversed, the ability to repress transcription is lost[10,11]. While the N-terminal portion of the 183-residue L5 repressor has a predicted HTH DNA-binding motif, bioinformatics and structure prediction programs fail to provide an understanding of the C-terminal region of this protein.

A recent report documented the successful use of mycobacteriophages for the treatment of a drug-resistant *Mycobacterium abscessus* infection, providing further support for the use of phage therapy in the treatment of multi-drug-resistant bacteria[15]. Two of the three phages used to treat the *Mycobacterium abscessus* infection were temperate, and these phages were genetically altered to remove their *repressors*, leading to lytic replication[15]. Additionally, recent work has also shown promise in the use of mycobacteriophages for the treatment of tuberculosis[16]. Given this therapeutic potential, a more detailed understanding of mycobacteriophage genetic regulation, including structures and functions of mycobacteriophage repressors, is needed.

Here we present the X-ray crystal structure of a mycobacteriophage repressor from the cluster A bacteriophage TipsytheTRex bound to DNA. The TipsytheTRex repressor shares 98% protein sequence identity to the L5 repressor, and its genome contains the same consensus operator/stoperator sites observed in L5 (Supplementary Fig. 1). The structure reveals that the repressor monomer contains two DNA-binding domains, an N-terminal HTH and a second domain that is a variation of the HTH motif. The crystal structure is supported by small-angle X-ray scattering solution studies, and we utilize immunity and DNA-binding assays to identify residues critical for repressor function. Finally, we employ molecular dynamics to investigate protein conformational flexibility on and off DNA. Overall, our structure presents a mechanism for how a monomeric repressor can promote transcriptional silencing.

## Results and discussion

**TipsytheTRex immunity repressor structure.** The structure of the TipsytheTRex repressor: DNA complex was determined using single-wavelength anomalous diffraction (SAD) phasing. During initial model building and refinement, it proved challenging to maintain proper geometry, and the agreement between the model and data was relatively poor. The structure was significantly improved when an AlphaFold[17] model of the repressor protein was included during building and refinement of the complex, the details of which are provided in the methods section.

The structure provides a clear explanation for how cluster A mycobacteriophage immunity repressors bind an asymmetric DNA sequence as a monomer[6,10], as it reveals that the protein engages the DNA sequence using two independent DNA-binding domains (Fig. 1). The first is an N-terminal HTH (residues 15–55) that was correctly predicted from the repressor sequence. The second domain (residues 75–181), which we have named the Stoperator domain, contains within it a fold that is a variation of the HTH motif (residues 75–118). A search for structural homologs of the Stoperator domain using DALI[18] returns no proteins with structural similarity. The DNA adopts a traditional B-form structure, with the HTH and Stoperator binding at successive openings of the major groove on the same face of the DNA in a manner similar to a cI dimer[8] (Fig. 1a–c). A small helix, which we have termed the helix bridge (residues 56–74), sits above the minor groove and connects the two DNA-binding domains. As expected, the HTH, helix bridge, and Stoperator are all rich in positive charge at the DNA contact points, and a PISA analysis shows that protein:DNA interactions in the structure result in 1636 Å$^2$ of buried surface area[19]. It is possible that cluster A repressors have evolved two DNA-binding domains within one polypeptide so that it can achieve good site occupancy of the many operator/stoperator sites located throughout the genome. This would promote a more economical mode of binding as the repressor monomer works to inhibit both transcription initiation and elongation.

The DNA-binding motif within the Stoperator domain maintains helices 1 and 3 (labeled α5 and α6 in Fig. 1a, respectively) of a traditional tri-helical HTH fold[20]; however, it is missing the second helix of the HTH and instead contains a long loop that connects helices α5 and α6. This key difference is illustrated when the Stoperator DNA-binding motif is superimposed on the TipsytheTRex HTH domain (root-mean-square deviation (RMSD) of 1.85 Å over 29 residues (Fig. 1d)). Additionally, while in typical HTH domains only the third, recognition helix is inserted into the DNA major groove, in the Stoperator domain portions of both α5 and α6 are inserted into the major groove where they contact the DNA (Fig. 1a). Along with missing helix 2, this motif also contains a small $3_{10}$-helix

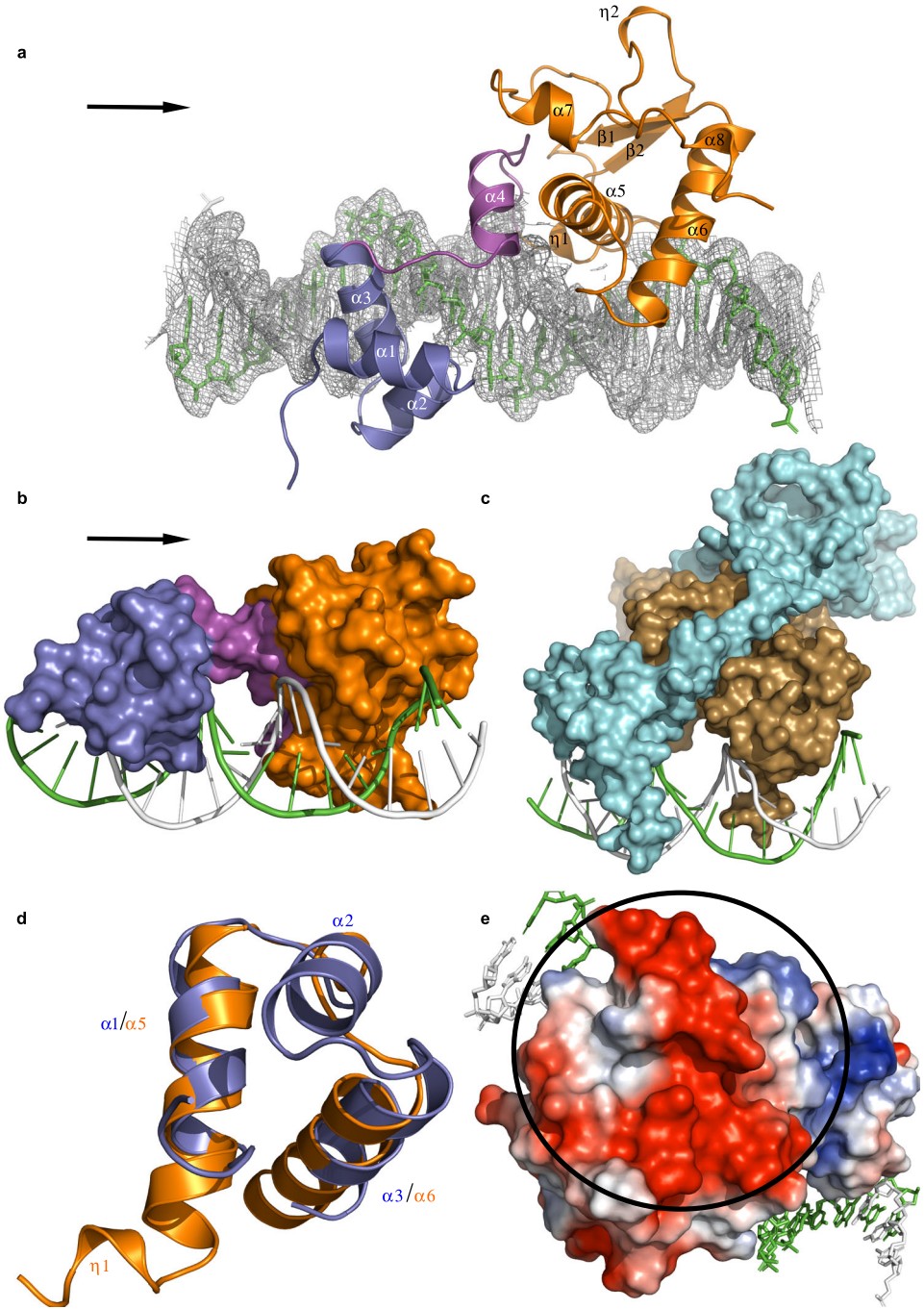

**Fig. 1 Overall structure of the repressor:DNA complex. a** Cartoon representation of the repressor bound to DNA. The protein monomer is colored by domain and includes the N-terminal HTH (residues 15–55, slate blue), the helix bridge (residues 56–74, magenta), and the Stoperator (residues 75–181, orange). The protein secondary structural elements are labeled as either alpha helices (α), beta strands (β), or 3$_{10}$ helices (η), and the two DNA strands are colored in green and white. $2F_o - F_c$ density, contoured at 1 sigma, is shown in gray for the DNA helix. The two DNA-binding domains engage the DNA via insertion into adjacent openings of the major groove, while the helix bridge serves as a linker between the two DNA-binding domains and lies above the minor groove. **b** The repressor is shown in surface view and is color-coded as in **a**. The arrows shown in **a**, **b** indicate the direction of transcription. **c** Surface view of the lambda cI dimer bound to DNA. This image was generated from coordinates 3BDN and shows the two monomers of the cI dimer colored in cyan and brown, with the two DNA strands colored in green and white. In cI, the HTH domain from each monomer of the dimer binds adjacent openings of the major groove. **d** Superposition of the repressor HTH (slate blue) and the DNA-binding motif of the Stoperator domain (orange). The Stoperator lacks the α2 helix of the HTH domain and also contains a small 3$_{10}$-helix (η1) at its N-terminus. **e** The region of the Stoperator domain that does not bind the DNA substrate is emphasized with a black circle. An electrostatic surface rendering (red: negative potential, blue: positive potential, white: neutral) reveals that this portion of the protein is acidic in nature.

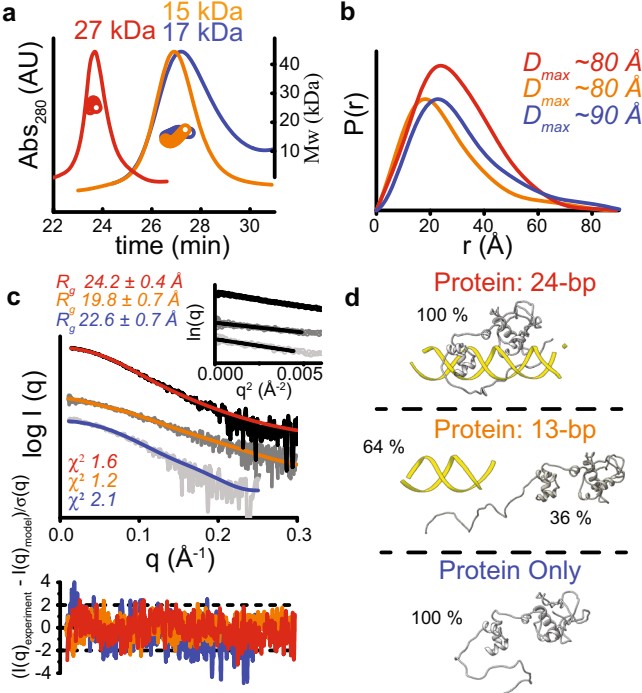

**Fig. 2 SAXS analysis of the repressor on and off DNA. a** SEC elution profiles for the repressor only (blue), repressor mixed with a 13-bp DNA (orange), and repressor mixed with a 24-bp DNA (red), along with masses calculated from MALS. The masses confirm that the repressor is a monomer in solution both on and off DNA. **b** $P(r)$ functions calculated from the experimental data that is shown in panel **c**, with the traces color-coded as in **a**. The distance $r$, in Angstroms (Å), on the $x$-axis where the $P(r)$ function approaches zero intensity represents the maximal dimension ($D_{max}$) for each sample. **c** Experimental SAXS curves are shown in gray/black along with the theoretical scattering profiles, fit-residuals, and $\chi^2$ values for the atomistic models of the protein:24-bp DNA (red trace), protein:13-bp DNA (orange trace), and protein only (blue trace) samples shown in panel **d**. The Guinier plots (inset) were used to calculate the radius of gyration values ($R_g$) for each sample. **d** Atomistic models derived from SAXS-fitting, with protein colored in gray and DNA in yellow. The percentages define how much each model contributes to the theoretical scattering profiles shown in panel **c**. Top panel: the protein:24-mer complex provides an excellent fit to the experimental data ($\chi^2 = 1.6$), confirming that the structure observed in the crystal matches the conformation of the complex in solution. The long loop at the N-terminus represents the first 14 residues of the protein as well as the His-tag that was disordered in the crystal structure. Middle panel: the protein:13-bp sample is best described by a mixture of free protein and free DNA in solution, in agreement with the SEC trace in panel **a**. Bottom panel: The free protein matches the SAXS data when it contains solvent-exposed N- and C-termini.

($\eta1$) at its N-terminus that contributes to DNA recognition in a manner similar to homeodomain proteins[21].

The C-terminal region of the Stoperator domain (residues 128–181) is the only portion of the protein that does not contact DNA, and this region appears to be quite flexible. A small $3_{10}$-helix ($\eta2$), two small helices ($\alpha7$ and $\alpha8$), and two beta strands ($\beta1$ and $\beta2$) make up the secondary structure in this region, with the rest composed of loops. An electrostatic potential surface rendering shows that this part of the protein is acidic in nature (Fig. 1e), in agreement with the high D and E content present in the C-terminus[11]. While the function of this region of the protein is currently unknown, we hypothesize that it may have some role in protein:protein interactions needed to regulate transcriptional silencing. These could be interactions with the host RNA

polymerase to halt transcription (discussed below) and/or interactions with other host or phage proteins during the lytic and lysogenic cycles. As no anti-repressor has been identified in these phage genomes[6,10,11], the latter proposal provides an intriguing explanation for how repression can be switched on and off in the host cell via protein contacts.

To confirm that the repressor is indeed a monomer, and that the structure observed in the crystal matches the conformation in solution, we analyzed free protein as well as protein mixed with both a DNA containing just the consensus sequence (13-bp) and a DNA that contains flanking nucleotides on each side of the consensus (24-bp) using small-angle X-ray scattering coupled with multi-angle light scattering in line with size-exclusion chromatography (SEC–SAXS–MALS)[22]. The predicted mass of the monomeric repressor is 23.6 kDa, and our MALS data, which shows a mass of 17 kDa, is consistent with a monomer in solution (Fig. 2a and Supplementary Table 1). When complexed with a 24-bp DNA, we see a shift in elution time that is most consistent with the formation of a monomeric repressor:DNA complex (Fig. 2a; predicted mass = 38.5 kDa; experimental mass = 27 kDa). Interestingly, when mixed with the 13-bp DNA, the SEC elution profile shows a mass similar to free protein, indicating that the consensus sequence alone is too small for the repressor to stably bind. The SAXS profiles plotted as normalized Kratky plots (Supplementary Fig. 2), together with the long tails observed in the $P(r)$ functions (Fig. 2b), suggest the presence of unfolded regions consistent with the long, unfolded N-terminal region that is missing in our crystal structure. Furthermore, the shift of Kratky plot maxima between the free protein and protein:DNA complex indicates an increased rigidity of the protein:DNA complex (Supplementary Fig. 2). A direct comparison of our crystal structure to the SAXS scattering profile for the protein:24-bp sample using FoXS[23,24] shows that the conformation of the protein:DNA complex in solution matches our structure (Fig. 2c, d, $\chi^2 = 1.6$). Additionally, the SAXS scattering profile of the protein:13-bp sample is best described by a mixture of free protein and free DNA ($\chi^2 = 1.2$), further supporting that the protein cannot stably bind just the consensus sequence. The protein-only SAXS scattering profile can be fit using a model that looks similar to the protein fold on DNA ($\chi^2 = 2.1$). However, the protein has reduced solubility off DNA, and therefore the quality of the protein-only data prevented a detailed modeling of the structure to the SAXS data. From the SEC–SAXS–MALS data, we can conclude that the repressor:DNA crystal structure matches the protein conformation in solution, and that more than the consensus sequence is required for formation of a stable protein:DNA complex.

**Protein:DNA interactions**. The TipsytheTRex repressor forms extensive interactions with both strands of the DNA substrate using the HTH, helix bridge, and Stoperator domains (Fig. 3a). While the helix bridge only contacts the DNA backbone, both the HTH and Stoperator domains form interactions with both the backbone and DNA bases. The majority of contacts occur within the 13-bp consensus sequence (underlined in Fig. 3a), although W50, Y55, and S111 interact with flanking nucleotides. The HTH domain has a canonical tri-helical structure[20], with the recognition helix ($\alpha3$ in Fig. 1) inserted into the major groove and providing base-specific contacts with the DNA. Figure 3b shows residues in the HTH that contribute to sequence specificity by directly contacting DNA bases. R45, which was previously predicted to be critical for DNA recognition[12], and Q46 sit at the start of the $\alpha3$ helix, and side by side they each interact with a guanine base on different strands of the duplex DNA. The only other residue in the HTH observed to contact a DNA base is

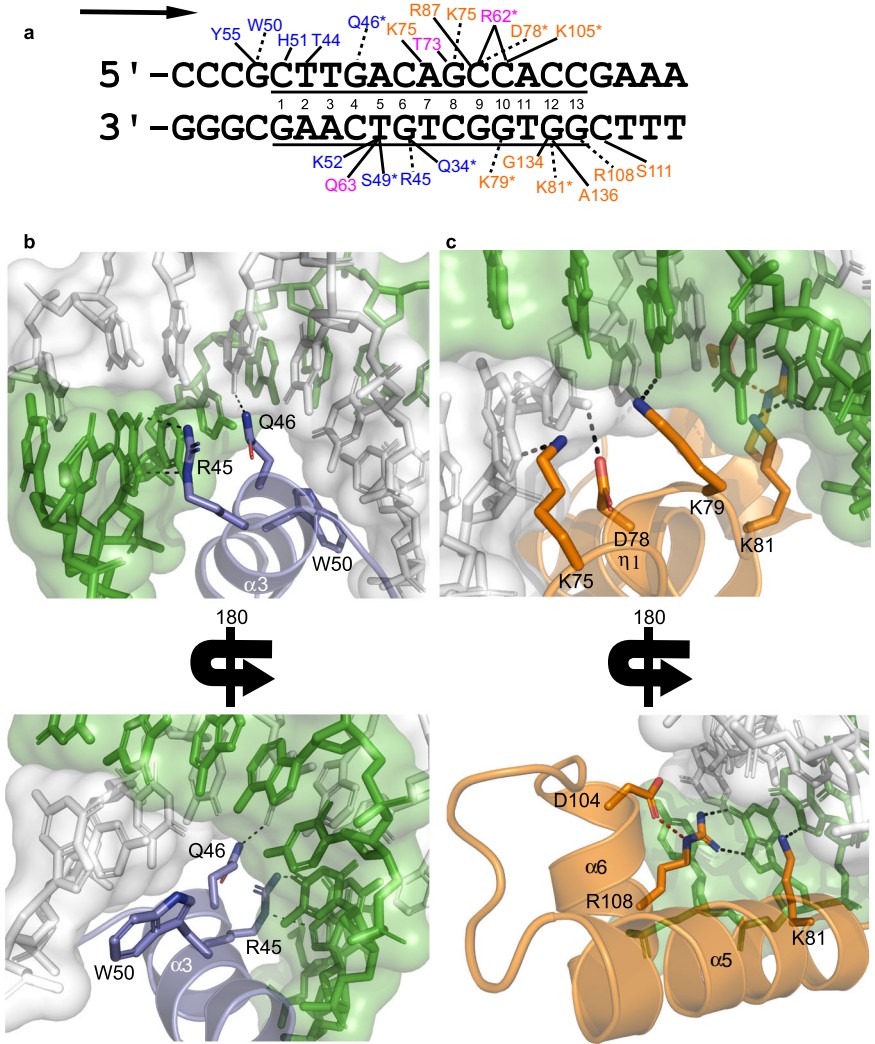

**Fig. 3 Repressor:DNA interactions. a** Double-stranded DNA sequence present in the crystal structure is shown, with the consensus sequence underlined. The numbers are present to identify each nucleotide of the consensus. Residues that contact the DNA are listed, with dashed lines indicating interactions with the DNA bases, while solid lines designate residues that contact the DNA backbone. All polar contacts shown are 3.2 Å or less, and an asterisk indicates contacts that are only observed in the higher resolution selenomethionine structure. Residues are color-coded as in Fig. 1a, and the arrow indictates the direction of transcription. **b** Interactions between R45, Q46, and W50 of the α3 helix in the HTH domain and DNA bases are shown. **c** Interactions between the Stoperator domain and DNA bases. At the beginning of this domain, the η1 3$_{10}$ helix properly positions K75, D78, and K79 to contact bases of the DNA. K81 sits at the base of the α5 helix. R108 in the α6 helix is properly positioned to bind DNA via an interaction (colored red) with D104. In both panels **b**, **c**, the protein and DNA are colored coded as in Fig. 1a.

W50, which stacks against a guanine located outside of the consensus sequence. This observation agrees with our SAXS results in that more than just the 13-bp consensus is needed for the repressor to bind DNA.

While the HTH and Stoperator domains bind DNA with roughly the same buried surface area (715 Å$^2$ and 751 Å$^2$ [19], respectively), the Stoperator DNA-binding motif contacts more DNA bases using residues in the η1 3$_{10}$-helix as well as in the α5 and α6 helices (Fig. 3c). The η1 3$_{10}$-helix at the start of this domain serves to position K75, D78, and K79 to interact with adenine, cytosine, and guanine bases in the consensus, respectively. K81 sits at the N-terminal end of the α5 helix where it contacts a guanine base, while R108 is positioned at the C-terminal end of the α6 helix and also contacts a guanine base. The R108 side chain also interacts with D104 present in α6 (indicated with red dashes in Fig. 3c), which helps to properly position the arginine to contact the DNA. Previous work reported a spontaneously arising L5 clear plaque mutant that contained an

R108L mutation in the L5 repressor and could no longer form lysogens[11], which suggests that the interaction between R108 and DNA observed in this structure is essential for formation of the repressor:DNA complex.

To identify which residues of the repressor described in Fig. 3b, c are critical for function, we performed immunity assays where *M. smegmatis mc²155* cells harboring an integration plasmid (pMH94) containing either the wild-type TipsytheTRex *repressor* plus its endogenous promoter, or mutations of residues that interact with DNA bases, were challenged with TipsytheTRex virus (Fig. 4a). Cells expressing the wild-type repressor reduced the efficiency of plating (EOP) of TipsytheTRex by greater than 4-logs as compared to vector-only cells. Results from this assay revealed that residues in the both the HTH and Stoperator domains are essential for function. In the HTH, both R45 and W50 are essential, with EOP's only ~3-fold lower than vector-only containing cells. Mutation of Q46 in the HTH leads to no loss of function. In the Stoperator domain, the greatest loss of

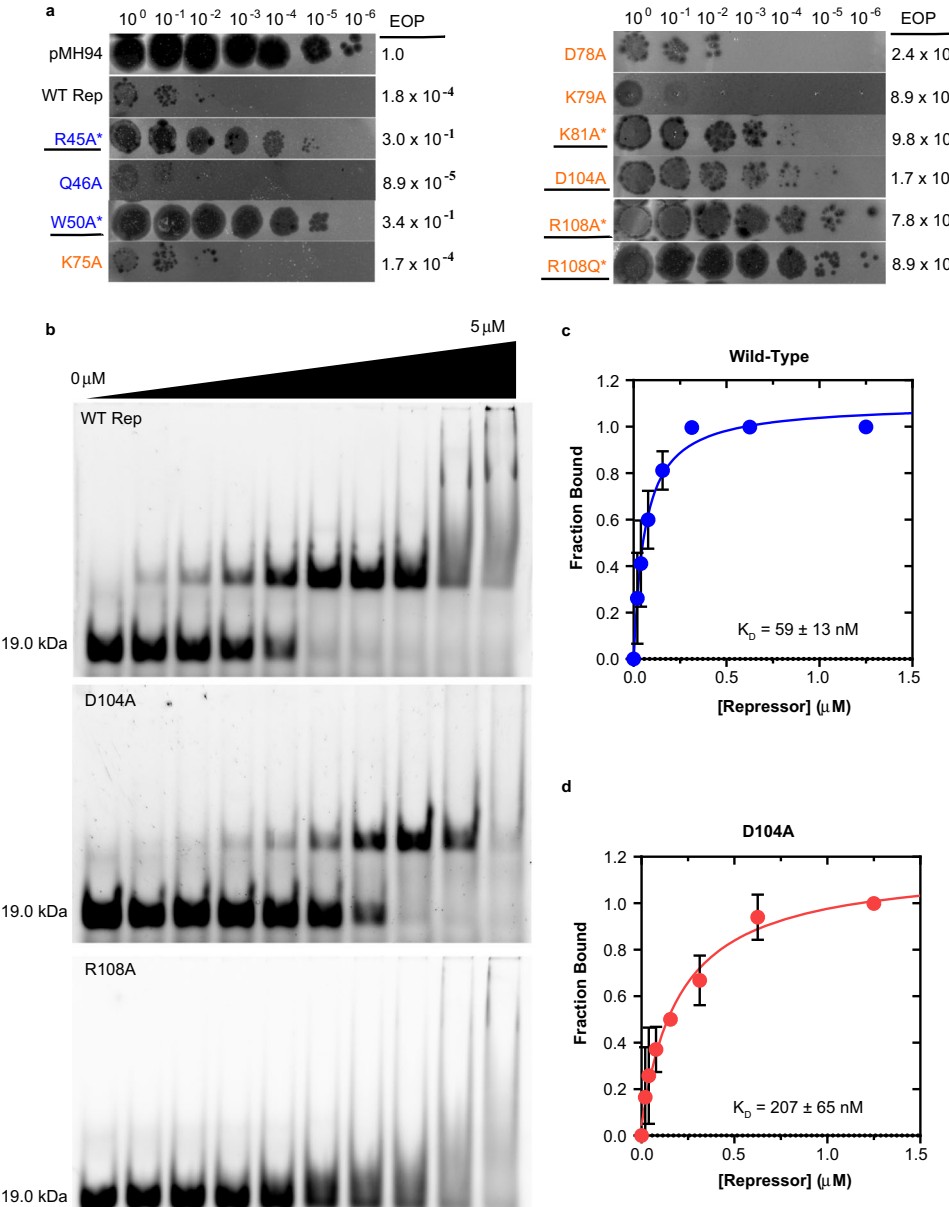

**Fig. 4 Residues critical for repressor function. a** TipsytheTRex virus was serially diluted and spotted onto top agar containing cells integrated with a single copy of the pMH94 empty vector, vector plus wild-type (WT) *repressor* and its endogenous promoter, or the indicated mutants. Spot titers and the efficiency of virus plating (EOP) were calculated as compared to vector-only containing cells. Mutants tested are color-coded to indicate whether they are from the HTH (blue) or Stoperator (orange) domains. Results shown are representative of three independent experiments. **b** Mutants underlined in panel **a** were tested for their ability to bind DNA using electrophoretic mobility shift assays. For each gel, fluorescein-labeled DNA was mixed with 0, 0.02, 0.04, 0.08, 0.16, 0.31, 0.63, 1.25, 2.5, or 5 μM protein. The mass of free DNA in the absence of protein (19.0 kDa) is indicated for each gel. A titration of the wild-type repressor (WT Rep) shows a shift in band size indicative of protein:DNA complex formation. The smearing present in the last two lanes represent non-specific complex formation present at high protein concentrations. While the D104A mutant retained the ability to bind DNA, the R108A mutant has lost the ability to form a specific complex with the DNA substrate. All mutants with an asterisk in panel **a** showed DNA-binding behavior similar to R108A (see Supplementary Fig. 6). This experiment was performed in three independent experiments, with similar results. **c, d** Both the wild-type (panel **c**, blue trace) and D104A (panel **d**, red trace) repressors bind the DNA ligand, with the D104A mutant displaying an ~3.5-fold weaker DNA-binding affinity as compared to wild-type. Plotted are the mean and standard deviation values calculated from three independent experiments. Source data are provided in the Source Data file.

function occurs with mutation of R108, with EOP's indistinguishable from vector-only containing cells. We see a complete loss of function when R108 is mutated to either an alanine or a glutamine, which suggests that the charge of the arginine is critical for function. Additionally, the D104A mutant also results in a loss of function, indicating that the D104:R108 interaction

observed in Fig. 3c is essential. We observe an intermediate phenotype with the K81A mutant, while residues in or near the η1 $3_{10}$-helix (K75, D78, and K79) show repression levels similar to wild-type.

Although the wild-type repressor strongly inhibited TipsytheTRex infection, some individual plaques were visible. To

determine if these plaques were caused by repressor escape mutant (REM) viruses, a full plaque assay was performed using wild-type repressor expressing cells. Five random plaques were picked, and all five were found to have EOPs close to 1 (Supplementary Fig. 3), indicating they were indeed REMs. Strikingly, resistance was conferred to each REM by the acquisition of only a single point mutation as compared to the wild-type stock (Supplementary Fig. 3). Four contained a nonpolar to polar mutation in either the fourth to last (M142T) or last (A145S) residue in gene product 74, which has no known function and is located adjacent to the *repressor*. The point mutation in the fifth REM resulted in a premature stop codon early in the *repressor* (E36Stop). Previous analyses of L5 spontaneous clear plaque mutants[11] and assorted cluster A repressor and lysogen escape mutants[6] revealed that mutations commonly occurred in the *repressor* as deletions, premature truncations, or frame shifts, as opposed to mutations in repressor DNA-binding sites. Mutations outside of the repressor have only rarely been observed, but one L5 mutant was reported to also contain a nonpolar to charged (A145E) mutation in the last amino acid of the gene upstream of the *repressor*[6]. Unlike most temperate bacteriophages, cluster A prophages are not strongly inducible by DNA damaging agents[9], and it is currently unknown if these lysogens are induced by other known mechanisms, such as quorum sensing[25], or by some novel route. While the role of gene product 74 in the regulation of lysogeny is currently unknown, the observation that mutation of this protein results in lytic growth points toward a critical function in the lysogenic switch. While AlphaFold predicts that the gene product 74 structure contains an α-β-α sandwich domain architecture (Supplementary Fig. 3), a DALI search to identify possible functions does not yield immediately actionable results. Additional work will be needed to elucidate gene product 74 function as it relates to repressor escape mutations.

To confirm that the loss of function observed in the immunity assays are due to defects in DNA binding, we monitored DNA-binding activity of mutants that displayed either intermediate or essential phenotypes from the results in Fig. 4a. Wild-type repressor efficiently binds a fluorescein-labeled 30-bp DNA substrate that contains the consensus motif (Fig. 4b, c). The binding of this sequence is specific and requires more than just the consensus, as the repressor fails to bind both a random 30-bp DNA that lacks the consensus as well as a 13-bp DNA that contains just the consensus motif (Supplementary Fig. 4). The lack of binding to the 13-bp DNA further supports our SAXS data and crystal structure. The wild-type repressor binds DNA with a $K_D$ of $59 \pm 13$ nM (Fig. 4c and Supplementary Fig. 5), which agrees well with the DNA-binding affinity reported for the L5 repressor[10].

All mutants that showed a loss of function in the immunity assays could be expressed and purified in a manner similar to the wild-type protein, and DNA-binding experiments confirm that R45A, W50A, K81A, R108A, and R108Q are all defective in DNA binding (Fig. 4b and Supplementary Fig. 6). These results provide support that the phenotypes observed for these mutants in the immunity assays are due to a loss of DNA interactions. D104A was the only mutant tested that retained specific DNA-binding activity, with ~3.5-fold weaker binding as compared to wild-type ($K_D = 207 \pm 65$ nM, Fig. 4d and Supplementary Fig. 5). The exact role of D104 in repressor function is unknown, but we predict that it is important for the proper positioning of R108.

Using the MEME suite[26], we identified 26 operator/stoperator sites within the TipsytheTRex genome that bear the same overall consensus sequence as L5 (Supplementary Fig. 7). An analysis of the various nucleotide substitutions in the 26 different sequences reveals important observations that correlate well with the

structure, immunity assay, and DNA-binding results. All sites within the consensus that interact with amino acid side chains that displayed a phenotype in our immunity assay (R45, K81, and R108) are 100% conserved in all sequences in both TipsytheTRex and L5, while those that did not show a phenotype engage the consensus where one or more substitutions are present. The greatest variation among the 26 sequences is observed in the middle of the consensus at positions 8 and 9, which interact with the side chains of K75 and D78. The K75A and D78A mutations showed no phenotype in our immunity assay, and from this we predict that this region can be highly variable without significantly impacting repressor function. Overall, the results show that the recognition sequence is well conserved in the 26 different sites, with residues that showed the greatest phenotype overlapping areas of highest conservation in the consensus.

**DNA binding occurs only when the HTH and stoperator domains are covalently linked**. To learn more about the contributions of the HTH and Stoperator domains in DNA binding, we expressed and purified the individual HTH and Stoperator domains and tested their DNA-binding abilities both alone and when mixed with the other domain. Our results show no specific DNA binding for the HTH alone, the Stoperator alone, or when the two proteins are mixed to mimic the full-length repressor (Supplementary Fig. 8). From these results, we conclude that the HTH and Stoperator domains must be fused into a single polypeptide to efficiently bind DNA. Such a requirement brings up an intriguing possibility for how repressor function may be regulated in order to switch from the lysogenic to the lytic cycle. Proteolytic cleavage of the helix bridge linker, which would separate the HTH and Stoperator domains, would be analogous to the lambda and P22 repressors that are inactivated when cleaved between the N-terminal HTH and C-terminal dimerization domains in response to recA[27]. Indeed, limited proteolysis studies on the L1 mycobacteriophage repressor (100% protein sequence identity to L5) revealed that the protein was preferentially cleaved by both trypsin and chymotrypsin at the helix bridge[14].

Given the lack of DNA binding observed for the individual domains, we next performed in silico molecular dynamics (MD) simulations with apo and DNA-bound TipsytheTRex repressor to learn more about communication between the HTH and Stoperator domains during DNA binding. Each MD trajectory was first analyzed to address whether conformational motions depend on DNA binding. Figure 5a, b present all correlated motions between the HTH and the Stoperator domain when in the apo or DNA-bound states. Inspection of Fig. 5a reveals few correlations between the HTH and Stoperator domains for the apo repressor. The few correlations that exist link motions between K81-D88 in the α5 helix of the Stoperator domain and T35-Y41in the α2 helix of the HTH domain. This contrasts the strong correlations observed in Fig. 5b for conditions with DNA bound, where increased correlated motions are expected for DNA-bound structures due to the sharing of a common DNA-lattice.

To understand condition-dependent conformational dynamics, each MD trajectory was subjected to Principal Component Analysis (PCA) to dissect all individual conformational motions contributing to the essential dynamics of the system. Figure 5c–h provide the results of PCA from conditions with the TipsytheTRex repressor structure either in the apo or DNA-bound states. For conditions lacking DNA (Fig. 5c), three principal components describe ~70% of system variance, where Principal Component 1 (PC1) and Principal Component 2 (PC2) each contribute 36% and 25%, respectively. Structural motions extracted for PC1 apo simulations indicate the dominant

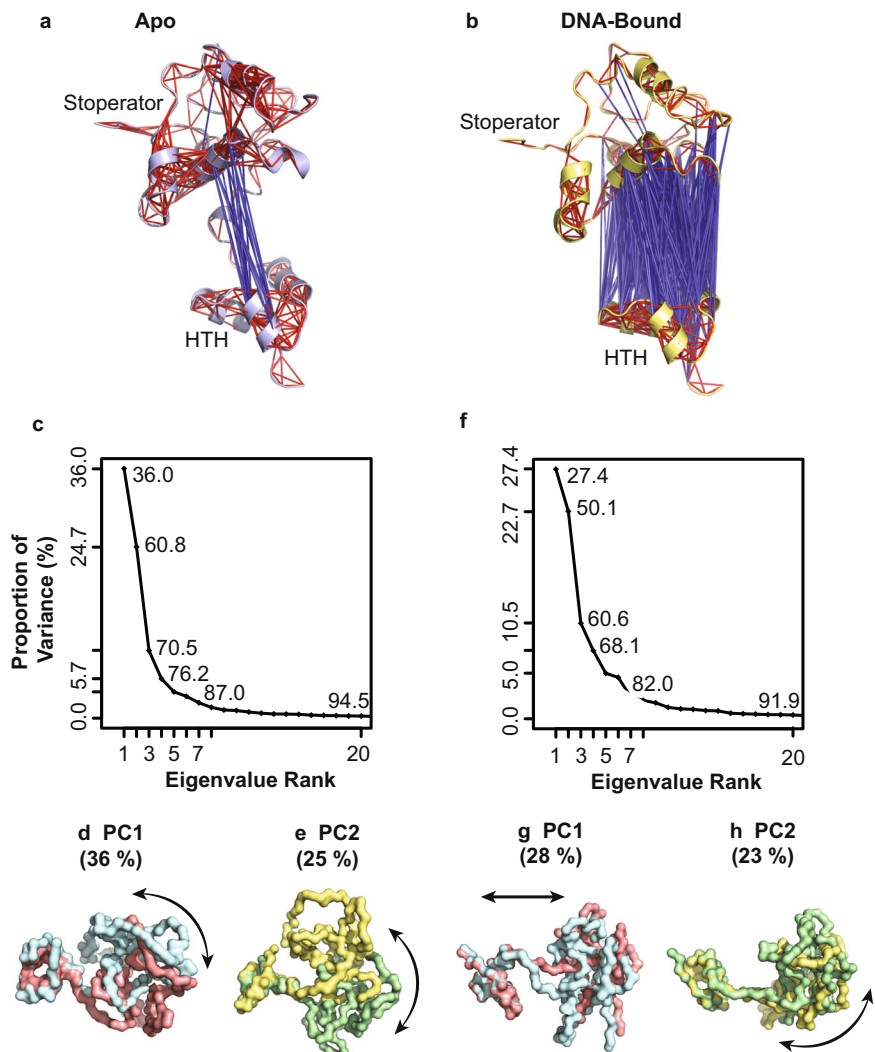

**Fig. 5 DNA binding promotes altered conformational dynamics. a, b** Cross-correlation analysis indicates a combination of positively (blue rods) and negatively (red rods) correlated conformational motions for the apo (**a**, slate blue) versus DNA-bound (**b**, yellow) TipsytheTRex repressor structure. All structural representations in **a, b** were prepared using Pymol 2.4.1 in ribbon view. **c** A Scree plot illustrates the contribution of individual principal components to overall system variance as a percentage based on MD trajectories generated using the apo repressor structure. *y*-axis labeling highlights the contribution of major principal components to system variance. Internal plot labeling presents total variance captured with each additional principal component. **d, e** Structural representations depict the conformational motions captured in Principal Components 1 (PC1, **d**, pink/cyan) and 2 (PC2, **e**, green/yellow). **f** A Scree plot illustrates the contribution of individual principal components to overall DNA-bound system variance as a percentage. **g, h** Structural representations depict the conformational motions captured in Principal Components 1 (PC1, **g**, pink/cyan) and 2 (PC2, **h**, green/yellow). All principal component structural representations are shown in surface view with the direction of motion indicated by an arrow.

motion to be a correlated rotation of the HTH and Stoperator domains about axes perpendicular to one another (Fig. 5d and Supplementary Video 1). In contrast, extracted PC2 structural motions reveal a pronounced transition between open and closed motions akin to those of a clamshell (Fig. 5e and Supplementary Video 2). Structural motions extracted for PC1 and PC2 each involve interactions that bridge the HTH and Stoperator domains, leading to transient adoption of a closed conformation of the repressor. The adoption of a closed repressor may allow for physiologic regulation of DNA-binding activities.

DNA binding to the repressor significantly impacts the range of conformational freedom experienced by the protein. In contrast to apo conditions, four principal components contribute ~70% of system variance when DNA is bound. PC1 and PC2 each contribute 28% and 23%, respectively (Fig. 5f). Inspection of the structural motions represented by PC1 indicates that the HTH and Stoperator domains undergo translations that effectively

involve extension and contraction of the helix bridge linker between them (Fig. 5g and Supplementary Video 3). PC1 does not describe significant rotational motions for either domain. In contrast, PC2 involves subtle rotations of both domains in perpendicular planes to one another (Fig. 5h and Supplementary Video 4). Despite the additional principal components necessary to describe the conformational dynamics of the DNA-bound repressor, the motions themselves are not as significant in magnitude as for the apo structure. That is to say, DNA binding drives significant stabilization of the repressor conformational motions such that multiple smaller motions contribute to overall movements, which contrasts the apo structure where 1-2 large motions drive the majority of system variance.

**DNA-binding energies are different for each domain.** To understand the thermodynamic contributions of the HTH and

**a**

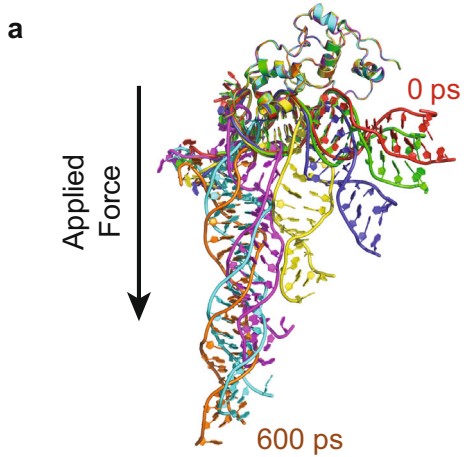

Applied Force

0 ps

600 ps

**Fig. 6 DNA-binding dissociation energies are different for the HTH and Stoperator domains.** Steered molecular dynamics techniques were employed alongside umbrella sampling methods to simulate DNA dissociation from full-length repressor. **a** Frames extracted along the COM pulling trajectory are superimposed to highlight dissociation path. All structural representations were prepared using Pymol 2.4.1 in ribbon view with frames corresponding to $t = 0, 100, 200, 300, 400, 500,$ and 600 ps colored as red, green, blue, yellow, magenta, cyan, and orange, respectively. The TipsytheTRex repressor protein structure is position restrained, while DNA is sequentially pulled along a defined path by application of a static force vector. The resulting SMD trajectories were then subjected to umbrella sampling techniques to calculate Potential of Mean Force (PMF). The amplitude of a resulting plot of PMF versus distance between protein and DNA centers-of-mass yields an estimate of the dissociation free energy (**b**). Adequate sampling was confirmed by weighted histogram analysis (**c**) and error estimation was obtained by bootstrap methods ($n = 200$). Figure labeling is present to indicate binding dissociation energy value, $\Delta G$.

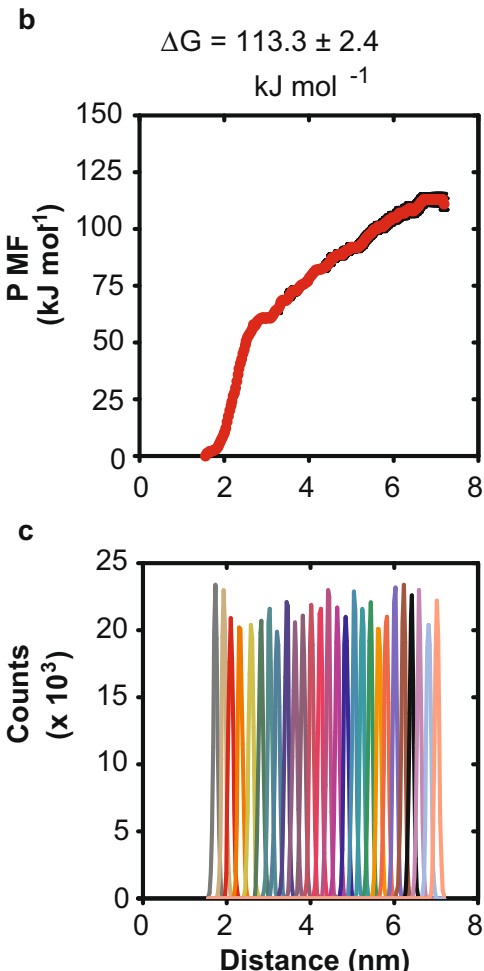

**b**

$\Delta G = 113.3 \pm 2.4$ kJ mol$^{-1}$

**c**

qualitative observations suggest that each binding domain interacts with DNA with different dissociation equilibrium constants.

The resulting plot of Potential of Mean Force (PMF) versus COM distance presented in Fig. 6b exhibits biphasic behavior such that a terminal plateau is observed when COM distances are greater than ~6 nm. The qualitative observation of non-simultaneous dissociation of HTH and Stoperator domains (Fig. 6a) and biphasic PMF curve (Fig. 6b) allow for prediction of individual domain binding dissociation energies based on the difference between PMF maximum and minimum for each phase[28–30]. The rationale for this statement is based on the observation of local maxima in the PMF curve that correspond to simulation states representative of step-wise domain dissociation from the initially bound DNA. An apparent local maximum is observed in Fig. 6b at COM distance equal to ~3 nm. Inspection of the corresponding extracted SMD trajectory frames reveal this local maximum to correspond to the dissociation of the Stoperator domain. By treating this as a local maximum, the $\Delta G_{\text{stoperator}}$ can be calculated as $60.7 \pm 0.3$ kJ mol$^{-1}$ and assigned as the Stoperator-binding dissociation energy. All subsequent COM distances sampled in Fig. 6b are representative of only HTH:DNA-binding interactions since the Stoperator domain is no longer DNA-bound. Moreover, qualitative inspection of extracted SMD trajectory frames derived from the terminal maximum at COM distances greater than 6 nm correspond to HTH domain dissociation. For this reason, the overall amplitude equal to $113.3 \pm 2.4$ kJ mol$^{-1}$ must represent the dissociation binding energy for the N-terminal HTH domain. Therefore, the resolved biphasic behavior presented in Fig. 6b allows for the estimation of HTH and Stoperator dissociation energies as $113.3 \pm 2.4$ and $60.7 \pm 0.3$ kJ mol$^{-1}$, respectively, occurring via step-wise events. This two-fold difference in binding energies calculated for each DNA-binding domain predicts step-wise-binding events that would facilitate HTH binding followed by Stoperator domain binding.

Stoperator domains to DNA binding, we applied a steered molecular dynamics (SMD) technique known as center-of-mass pulling (COM pulling) to induce dissociation of DNA from the repressor, allowing for calculation of dissociation free energy values. Figure 6a provides a visual overview of DNA dissociation with applied force and highlights that the HTH and Stoperator domains do not simultaneously release the initially bound DNA. The Stoperator domain is observed to dissociate from the DNA first, followed by the HTH domain. After a 600 ps force application, the DNA is fully dissociated from the repressor. These

**Model for transcriptional silencing in the host**. Cluster A bacteriophages do not encode their own RNA polymerase and must rely on the host for transcription[9]. A fundamental question is how does the repressor halt transcription by the host RNA polymerase? At operator sites in promoter regions, binding of the repressor would simply block the ability of the polymerase to bind the promoter to initiate transcription. However, how does this work at stoperator sites that halt transcription elongation? The model in Fig. 7a shows that the RNA polymerase would encounter the HTH domain of the repressor first, and one

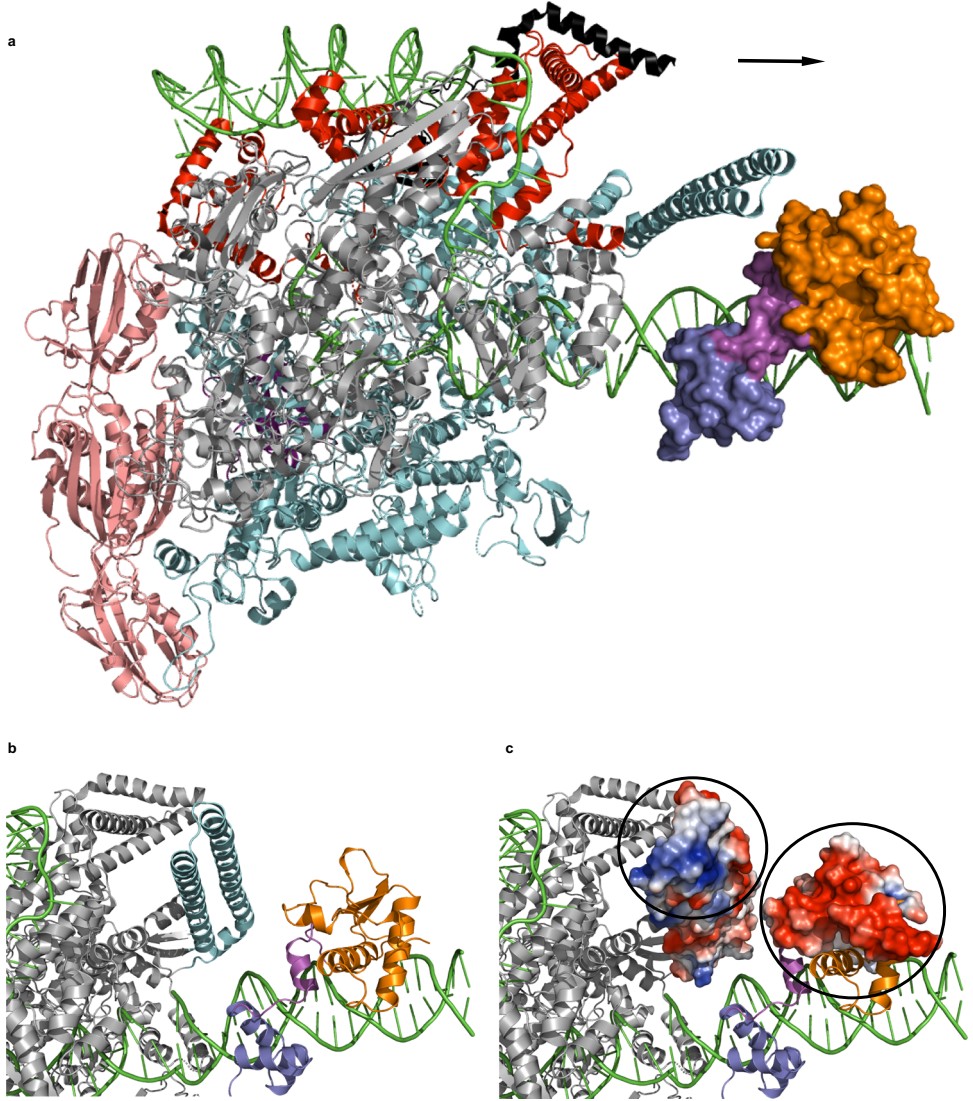

**Fig. 7 Model for transcriptional silencing in cluster A mycobacteriophages. a** The *Mycobacterium smegmatis* RNA polymerase (PDB 5VI5) is colored as follows: α subunits: pink, β subunit: gray, β' subunit: cyan, ω subunit: dark purple, σ^A subunit: red, and RNA polymerase-binding protein RBPA: black. The TipsytheTRex repressor is shown in surface view and colored as in Fig. 1a, and it has been rotated forward by ~90° relative to the orientation observed in Fig. 1. All nucleic acid in the figure is colored green, and the direction of transcription is indicated with an arrow. The repressor may inhibit transcription elongation by either serving as a steric block or by halting the polymerase via protein:protein interactions. **b** Zoomed in view showing the position of the RNA polymerase β' insert (colored cyan) relative to the C-terminal region of the Stoperator domain of the repressor (colored orange). The remaining regions of the RNA polymerase are colored in gray for clarity. **c** Same view as in panel **b** but with electrostatic potentials shown (red: negative potential, blue: positive potential, white: neutral) for the RNA polymerase β' insert and C-terminal region of the Stoperator domain of the repressor. Circles are drawn around areas of positive and negative charge for the β' insert and C-terminal region of the repressor, respectively.

hypothesis is that the repressor halts elongation simply by acting as a steric block that impedes transcription. However, previous work revealed that reversing the orientation of repressor asymmetric-binding sites results in failure to repress transcription elongation, which argues against a simple steric block and indicates that the orientation of the N- and C-terminal domains of the repressor relative to the RNA polymerase is critical for this function[10]. An alternative hypothesis is that transcription is halted via communication between the host polymerase and the repressor, and there are features of both proteins that would allow for this communication to be specific. The *M. smegmatis* RNA polymerase contains an insert in the β' subunit (colored cyan in Fig. 7a, b) that is unique to actinobacteria, as well as an N-terminal segment of the σ^A protein (colored red in Fig. 7a) that is unique to mycobacteria[31]. Modeling of an RNA

polymerase:repressor complex positions the C-terminal portion of the repressor's Stoperator domain near the β' insert (Fig. 7). Interestingly, analysis of the electrostatic potential shows that a positive patch on the β' insert lies adjacent to the negative charge seen on the C-terminal end of the repressor (Fig. 7c), and such an electrostatic interaction network would be similar to the mode of binding observed between the lambda cI protein and the sigma subunit of the *E. coli* RNA polymerase during transcriptional activation[7,32]. The majority of the unique σ^A N-terminal segment was disordered in the RNA polymerase structure, so it is unclear if that portion of the protein would reach the repressor's C-terminus in this model.

An alignment of repressors from 336 cluster A mycobacteriophages (87 of which were grouped into an L5 clade) revealed much greater sequence diversity in the C-terminal Stoperator

domain than in the N-terminal HTH[6]. Intriguingly, when L5 lysogens were challenged with different bacteriophages in the L5 clade, complex immune phenotypes were observed that correlated with the diversity observed in the C-terminal Stoperator domain[6]. We hypothesize that while the two DNA-binding domains work together to halt transcription, it is the variations observed in the Stoperator domain that allow mycobacteriophages to evolve to escape superinfection immune mechanisms of other closely related mycobacteriophages. Future work will be needed to validate this model to show how temperate mycobacteriophages maintain lysogeny, switch between the lysogenic and lytic replication cycles, and evolve to evade superinfection immunity.

## Methods

**Oligonucleotides**. Deoxynucleic acids were purchased from Integrated DNA Technologies (IDT). All DNA substrates used for crystallization, SEC–SAXS–MALS, and DNA binding are provided in Supplementary Table 2. Double-stranded substrates were prepared by mixing the complements together in a 1:1 ratio in 10 mM Tris (pH 8.0), 50 mM NaCl, and 1 mM EDTA, heated to 95 °C, and then allowed to cool to room temperature in a water bath.

**Bacteriophage isolation, genomic sequencing, and virus propagation**. Bacteriophage TipsytheTRex was isolated from a soil sample on the campus of Western Carolina University (Cullowhee, NC) by student Brooke Burns in 2015. TipsytheTRex virus was plaque purified and its full genome sequenced, with Illumina Sequencing and genome assembly performed at the Pittsburgh Bacteriophage Institute. The TipsytheTRex genome was annotated by students at Western Carolina University and deposited in GenBank (accession number MF919536). For the current study, a plaque-picked stock of TipsytheTRex was grown from the original stock as described below. Plaque-picked TipsytheTRex and REM viral DNA was purified using phenol-chloroform extraction, followed by ethanol precipitation. Sequencing libraries were prepared using the NEBNext Ultra II FS DNA library kit. Libraries were pooled and sequenced using a MiSeq v2 300 cycle micro reagent kit on a MiSeq FGx instrument in RUO mode at Western Carolina University. 50–150 k reads (average 143 to 429x coverage) were used for de novo genome assemblies. Genomes were assembled and quality examined using the programs Newbler V2.9, Ace Util, and Consed V29[33]. The TipsytheTRex stock used in this study was found to contain two mutations (22,957T to A (minor tail protein S360R) and 44,843T to C (repressor protein T60A)), as compared to the GenBank entry.

TipsytheTRex virus was isolated and propagated using *Mycobacterium smegmatis* mc²155 high-frequency transformation strain cells (a kind gift from the Hatfull Laboratory). *Mycobacterium smegmatis* mc²155 cells were grown to saturation in Middlebrook 7H9 media containing 10% AD supplement, 1 mM CaCl₂, and 0.05% tween 80 at 37 °C. Cells were then diluted 1:100 and grown to saturation in Middlebrook 7H9/AD/CaCl₂ media lacking tween. For virus stock growth, serial dilutions of TipsytheTRex were mixed with 0.5 mL of cells and 4.5 mL Middlebrook 7H9 Top Agar, then poured onto Middlebrook 7H9 plates. Plates were incubated for 24 h at 37 °C. Webbed plates were then flooded with 8 mL of Phage Buffer (10 mM Tris, 10 mM MgSO₄, 68 mM NaCl, 1 mM CaCl₂, pH 7.5), and lysates were harvested and filtered using 0.22 μm PES syringe filters.

**Protein expression and purification**. The TipsytheTRex *repressor* was amplified from bacteriophage genomic DNA, then subcloned into the pET28a vector between NdeI and XhoI sites using the NEB HiFi DNA assembly kit. The resulting construct produces the repressor protein with an N-terminal His₆ tag. The repressor protein was expressed in *E. coli* BL21(DE3) cells by growing in LB broth at 37 °C, with shaking, until the OD₆₀₀ = 0.6–0.8. Cells were cooled on ice, then induced with 1 mM Isopropyl β-d-1-thiogalactopyranoside at 16 °C with shaking overnight. Cells were harvested by centrifugation and resuspended in 50 mM Tris pH 8.0, 0.5 M NaCl, 0.1 mM EDTA, 10% glycerol, and 0.1 mg mL⁻¹ lysozyme. Cells were lysed by sonication, followed by a 1-h centrifugation at 34,541 × g at 4 °C. The resulting supernatant was loaded on a 5 mL HiTrap nickel column (GE) equilibrated in 50 mM Tris pH 8.0, 0.5 M NaCl, 0.1 mM EDTA, 10% glycerol, and 10 mM imidazole. The column was washed with 40 mM imidazole, and the protein eluted by increasing the imidazole to 250 mM. Fractions containing the repressor protein were identified by SDS-PAGE, pooled, diluted to 0.1 M NaCl in 20 mM Tris pH 7.5, 0.5 mM EDTA, 5% glycerol, and 1 mM betamercaptoethanol buffer, and loaded on a 5 mL HiTrap heparin column (GE). The protein was eluted from heparin using a 0.1–1.5 M NaCl gradient over 200 min. Fractions containing the repressor protein were identified by SDS-PAGE, pooled, then dialyzed against 2 L of buffer containing 20 mM Tris pH 7.5, 0.5 M NaCl, 0.5 mM EDTA, 5% glycerol, and 0.5 mM dithiothreitol. The protein was concentrated, then flash frozen in liquid nitrogen for storage at −80 °C. For experimental phasing, selenomethionine-derivatized repressor protein was prepared in BL21(DE3) cells using published methods[34] and purified as described above.

**X-ray structure determination and refinement**. To prepare for crystallization, 250 μM repressor (5.4 mg mL⁻¹) was mixed with an equal concentration of double-stranded DNA substrate and allowed to incubate at room temperature for 30 min. During this time the solution turned cloudy, but we observed that the addition of 40 mM MgCl₂ caused the solution to clear. Crystals of both native and selenomethionine-derivatized repressor:DNA complex were obtained by hanging-drop vapor diffusion at 22 °C by mixing 1 μL protein with 1 μL reservoir solution placed over a 500 μL reservoir solution containing 5–12% PEG 8000, 0.1–0.3 M CaCl₂, and 0.1 M HEPES pH 7.5. Crystals appeared in 3-4 days and grew to full size by one week. No repressor crystals were observed in the absence of DNA. To prepare native crystals for freezing, a solution containing the reservoir plus 30% ethylene glycol was prepared and slowly added directly to the drop containing the crystals. After equilibration, crystals were then quickly transferred to the reservoir 30% ethylene glycol solution, then flash frozen by plunging into liquid nitrogen. A similar procedure was used for selenomethionine-derivatized crystals except that 30% glycerol was used as the cryoprotectant. X-ray data were collected at wavelengths of 1.11608 Å and 0.979690 Å for the native and selenomethionine datasets, respectively, at beamline 8.3.1 at the Advanced Light Source on a Dectris Pilatus3 S 6 M detector. The native and selenomethionine datasets were processed using XDS V20190315[35] and HKL3000 V721.3[36].

Structure solution and refinement were performed using a combination of programs from Phenix (V1.19.2-4158 and V1.20-4459) and CCP4i (V1.0.2). The structure of the repressor:DNA complex was determined from the selenomethionine dataset by single-wavelength anomalous diffraction (SAD) phasing using a combination of the programs Crank2[37], Phenix AutoSol[38], and Autobuild[39]. Crank2 was able to build the majority of the protein model, with an $R_{work}$ and $R_{free}$ of 40.0% and 42.5%, respectively, while Phenix was able to build a portion of the DNA. The DNA from the Phenix solution was combined with the Crank2 protein model, and the missing protein and DNA components were manually built. Manual structure building was followed by xyz coordinate, real space, and individual B-factor refinement using the program phenix.refine to yield a preliminary model of the selenomethionine structure. This structure was then used as a search model in molecular replacement using the native dataset, followed by manual rebuilding and refinement. The preliminary models of the native and selenomethionine structures differed in the region containing residues 111–146, a region that was difficult to interpret for both structures. The model of the native structure was missing residues 118–128 and 139–147, and the model of the selenomethionine structure was missing residues 115–140. At this stage it proved challenging to maintain proper geometry during refinement, and the best $R_{work}$/$R_{free}$ values that were obtained for the native and selenomethionine models were 26.9/29.5% and 32.9/38.2%, respectively.

To improve the structures, we took advantage of the recent advances in structure prediction[17] to generate a model of the repressor alone. We used the Phenix version of the ColabFold Google Colabs notebook[40] available at (https://colab.research.google.com/github/phenix-project/Colabs/blob/main/alphafold2/AlphaFold2.ipynb) and obtained an AlphaFold model with an average plDDT (confidence) of 87.8 (a high confidence level). The AlphaFold model was similar to the preliminary model of the native structure that had been automatically and hand-built, but importantly it filled in the two gaps in the model that had remained unmodelled up to that point in a way that agreed closely with the density map of the native structure. The same AlphaFold model yielded an interpretation of the selenomethionine structure that agreed with the density map and that was different from the previous interpretation of residues 140–146 in the selenomethionine structure.

The AlphaFold model suggested a substantial change in interpretation of part of the repressor structure, so we stepped back and set out to obtain a single electron density map that could represent both the native and selenomethionine structures and that had no bias from the AlphaFold model. To do this, we first used the Phenix AutoSol software to calculate a density map for the selenomethionine structure using the anomalous differences in that dataset and including the DNA from the preliminary structure of the native complex in the phase calculations. Then we carried out multi-crystal density modification with the Phenix multi_crystal_average software using the resulting density map and the corresponding data from the selenomethionine dataset, combined with the measured amplitudes from the native dataset. The AlphaFold model was then docked directly into the resulting map using the Phenix dock_in_map software. The fit to this map was good even in the somewhat unclear region containing residues 111–146. The docked model was then combined with the DNA structure from the preliminary native complex, and this complex was refined against the native data with the Phenix real_space_refine software. In parallel, the AlphaFold model was used to replace the model of the repressor in the preliminary selenomethionine structure, and that complex was refined against the selenomethionine data.

The structures were further improved by manual rebuilding and refinement using the programs PDB-REDO[41] and REFMAC[42]. For the final stages of refinement, xyz coordinate, real space, TLS, and individual B-factor refinement were performed using the program phenix.refine, with both an ideal B-form 21-bp DNA and the repressor AlphaFold model used as reference model restraints. The final refined models contain an $R_{work}$/$R_{free}$ of 20.9/24.8% and 19.9/24.5% for the native and selenomethionine structures, respectively, with good geometry. The His-tag and first fourteen residues of the N-terminus, as well as the C-terminal two

**Table 1 Data collection and refinement statistics for the TipsytheTRex crystal structures.**

|  | Selenomethionine | Native |
|---|---|---|
| Data collection |  |  |
| Space group | C2 | C2 |
| Cell dimensions |  |  |
| a, b, c (Å) | 131.27, 44.17, 89.45 | 132.49 43.51 89.28 |
| α, β, γ (°) | 90, 102.90, 90 | 90, 102.26, 90 |
| Resolution (Å) | 46.83-2.79 (2.89-2.79)[a] | 47.39-3.13 (3.24-3.13) |
| Wavelength (Å) | 0.979690 | 1.11608 |
| Total Reflections | 20441 (910) | 17955 (1739) |
| Unique Reflections | 10937 (521) | 9005 (666) |
| CC 1/2 | 0.996 (0.739) | 0.999 (0.748) |
| CC[a] | 0.999 (0.922) | 1 (0.925) |
| $R_{merge}$ (%) | 5.97 (39.24) | 3.29 (43.17) |
| I/σ | 11.00 (1.71) | 12.20 (1.73) |
| Wilson B-factor | 65.90 | 92.15 |
| Completeness (%) | 85.37 (41.40) | 94.43 (75.20) |
| Redundancy | 1.9 (1.7) | 2.0 (2.0) |
| Refinement |  |  |
| No. reflections | 10888 (520) | 8542 (661) |
| $R_{work}/R_{free}$ (%) | 19.86/24.52 (35.84/36.83) | 20.87/24.76 (32.76/33.36) |
| No. atoms |  |  |
| Protein/DNA | 2257 | 2256 |
| Water | 0 | 0 |
| Average B-factors |  |  |
| Protein/DNA | 82.65 | 105.40 |
| Stereochemical ideality |  |  |
| RMS Bond lengths (Å) | 0.013 | 0.010 |
| RMS Bond angles (°) | 1.78 | 1.60 |
| ϕ,ψ most favored (%) | 95.76 | 95.76 |
| ϕ,ψ allowed (%) | 4.24 | 4.24 |
| ϕ,ψ outliers (%) | 0.00 | 0.00 |

[a]Values in parentheses are for highest-resolution shell.
Coordinates for the selenomethionine and native structures have been deposited in the Protein Data Bank as entries 7TZ1 [https://doi.org/10.2210/pdb7TZ1/pdb] and 7R6R [https://doi.org/10.2210/pdb7R6R/pdb], respectively.

residues were disordered in both structures and are not included in the final models. A summary of data collection and refinement statistics are provided in Table 1. Structure factors and coordinates have been deposited in the PDB as entries 7R6R and 7TZ1 for the native and selenomethionine structures, respectively.

**Small-angle X-ray scattering coupled with multi-angle light scattering in line with size-exclusion chromatography (SEC–SAXS–MALS).** For SEC–SAXS–MALS experiments, a 140 μL sample that contained either 270 μM repressor (5.8 mg mL⁻¹), or 270 μM repressor plus 235 μM DNA substrate, was prepared in 20 mM Tris pH 7.5, 0.5 M NaCl, and 1 mM DTT. SEC–SAXS–MALS data were collected at the ALS beamline 12.3.1 LBNL Berkeley, California[43]. The X-ray wavelength was set at λ = 1.127 Å and the sample-to-detector distance was 2100 mm resulting in scattering vectors, q, ranging from 0.01 Å⁻¹ to 0.4 Å⁻¹. The scattering vector is defined as $q = 4\pi sin\theta/\lambda$, where $2\theta$ is the scattering angle. All experiments were performed at 20 °C, with a SAXS flow cell directly coupled with an online Agilent 1260 Infinity HPLC system using a Shodex KW803 column. The column was equilibrated with running buffer as indicated above with a flow rate of 0.45 mL min⁻¹. 55 μL of each sample was run through the SEC and three-second X-ray exposures were collected continuously during a 35-min elution. The SAXS frames recorded prior to the protein elution peak were used to subtract all other frames. The subtracted frames were investigated by radius of gyration ($R_g$) derived by the Guinier approximation $I(q) = I(0) \exp(-q^2R_g^2/3)$, with the limits $qR_g < 1.5$[44]. The elution peak was mapped by comparing the integral of ratios to background and $R_g$ relative to the recorded frame using the program SCÅTTER V.e. Final merged SAXS profiles, derived by integrating multiple frames at the elution peak, were used for further analysis, including a Guinier plot to determine an aggregation free state. The program SCÅTTER was used to compute the $P(r)$ function. The distance r where $P(r)$ approaches zero intensity identifies the maximal dimension of the macromolecule ($D_{max}$). The eluent was subsequently split 3 to 1 between the SAXS line and a series of UV at 280 and 260 nm, MALS, quasi-elastic light scattering (QELS), and refractometer detectors. MALS experiments were performed using an 18-angle DAWN HELEOS II light scattering detector connected in tandem to an Optilab refractive index concentration detector (Wyatt Technology). System normalization and calibration was performed with BSA using a 45 μL sample at 7 mg mL⁻¹ in the same SEC running buffer and a dn/dc value of 0.19. The light scattering experiments were used to perform analytical scale chromatographic separations for molecular

mass determination of the main peaks in the SEC analysis. UV, MALS, and differential refractive index data was analyzed using Wyatt Astra 7 V7.1.48 software to monitor the homogeneity of the sample across the elution peak complementary to the above-mentioned SEC–SAXS signal validation.

**SAXS solution structural modeling.** Disordered regions of the protein that were missing in the crystal structure, which included the His-tag and first fourteen residues of the N-terminus, had to be added to the model for SAXS structural modeling using MODELLER V9.25[45]. The 24-bp and 13-bp DNA substrates were modeled using the 21-bp substrate present in the crystal structure. Minimal molecular dynamic (MD) simulations were performed on flexible regions of the protein using BILBOMD V2.0[46]. The experimental SAXS profiles were then compared to theoretical scattering curves generated from the apo protein and protein:DNA models using FoXS[23,24]. The SAXS data and atomistic models have been deposited in the SASBDB database as entries SASDMK3 (repressor only), SASDML3 (Repressor:24-bp DNA), and SASDMM3 (Repressor:13-bp DNA).

**Superinfection immunity assays and REM isolation.** The wild-type Tipsythe-TRex *repressor*, plus 257 bp of the upstream intergenic region that contains an endogenous promoter, was PCR-amplified from genomic DNA. The *repressor* plus intergenic was then cloned into the EcoRI site of the pMH94 integration shuttle-vector[47] and verified by sequencing. The pMH94 vector was a kind gift from the Hatfull laboratory. Site-directed mutagenesis of the pMH94 vector plus the *repressor* was used to generate the panel of repressor mutants. All mutant *repressor* sequences were verified via Sanger sequencing. Plasmid growth, cloning, and site-directed mutagenesis was performed using NEB5 alpha *E. coli* cells. PCR products and plasmids were purified using NEB Monarch Nucleic Acid Purification Kits.

For the superinfection immunity assays, purified pMH94 empty vector, or pMH94 plasmids containing TipsytheTRex *repressor* constructs, were electroporated into electrocompetent *Mycobacterium smegmatis* mc²155 cells and plated onto Middlebrook 7H9 plates containing 5 μg mL⁻¹ Kanamycin. Plates were incubated for 4 days at 37 °C. Bacterial colonies were then picked and grown to saturation in liquid culture as described above, with the addition of 5 μg mL⁻¹ Kanamycin. 0.5 mL of cells were then mixed with 4.5 mL Middlebrook 7H9 top agar (5 μg mL⁻¹ Kanamycin) and poured onto Middlebrook 7H9 plates (5 μg mL⁻¹ Kanamycin). A TipsytheTRex virus stock (10¹⁰PFU mL⁻¹) was then serially diluted in phage buffer, and 2 μL droplets were spotted in triplicate onto the top agar layer. Plates were incubated for 48 h at 37 °C. Spot titers and Efficiencies of Plating (Repressor Construct Spot Titer/pMH94 Empty Vector Spot Titer) were then calculated and averaged from three independent experiments for each sample. Uncropped images of all plates used for spot titer and Efficiencies of Plating calculations are provided in the Source Data document.

To isolate REMs, a full plaque assay was performed using cells expressing the TipsytheTRex repressor protein, as described above. Five plaques were picked and subjected to a second round of plaque purification on *Mycobacterium smegmatis* mc²155 cells, and stocks were grown as described above.

**DNA-binding assays.** Electrophoretic mobility shift assays (EMSA) were used to monitor repressor binding to DNA substrates. Site-directed mutagenesis of the pET28a vector plus the *repressor* was used to generate the panel of repressor mutants. All mutant *repressor* sequences were verified via Sanger sequencing. Mutant proteins were expressed using the same protocol described for the wild-type protein, and they were purified using Nickel column chromatography. For the wild-type repressor and all point mutations, 5 μM protein was mixed with 0.25 μM labeled DNA substrate in binding buffer that contained 20 mM Tris pH 7.0, 25 mM NaCl, 1 mM DTT, 100 μg mL⁻¹ BSA, and 5% glycerol. This complex was immediately serially diluted two-fold in a solution containing 0.25 μM labeled DNA substrate in binding buffer to monitor a protein concentration range that varied from 0.02–5 μM. Once diluted, the samples were incubated at room temperature for 30 min, then 10 μL of each sample was loaded on a 5% native acrylamide gel prepared in 1X TAE buffer. The gels were run at 65 V for 1 h at room temperature in 1x TAE buffer, then imaged for fluorescein fluorescence using a Bio-Rad ChemiDoc MP System. Non-specific DNA band migration was observed in the highest protein concentrations for all proteins, so for binding analysis only lanes that had DNA bands indicative of specific complex formation were quantified. Both free DNA and protein:DNA complex band intensities were quantified using the Image Lab V5.2.1 software. For the wild-type and D104A mutant, DNA-binding data were fit using a one-site binding model. Uncropped images of all gels used for DNA-binding analysis are provided in either the Source Data document or in the Supplementary Information file.

The individual *hth* and *stoperator* domains were PCR amplified from genomic DNA, then subcloned into the pET28a vector between NdeI and XhoI sites. The constructs were confirmed via Sanger sequencing. The proteins were expressed using the same protocol described for the wild-type protein, and they were purified using Nickel column chromatography. DNA-binding experiments with individual HTH and Stoperator proteins were carried out as described for the full-length proteins except that protein concentration ranges of both 0.02–5 μM and 0.1–25 μM were tested. All DNA-binding experiments were performed in triplicate.

**Identification of operator and stoperator sites**. To identify operator and sto-perator sites in the TipsytheTRex and L5 genomes, the TipsytheTRex and L5 genome fasta files were loaded into the MEME sever[26] and analyzed using the same parameters previously described for the analysis of other cluster A mycobacteriophage genomes, which are: site distribution of any number of repetitions, maximum of two motifs, motif length of 12 to 16 bp, a range of 10-50 sites, and derived from both strands[6].

**Equilibrium molecular dynamics**. Prior to use in molecular simulation software, a complete TipsytheTRex repressor protein chain was constructed by modeling and joining unresolved residues L116–P131 and R141-D146 using UCSF Chimera X[48]. The resulting model was then minimized by steepest descent in Gromacs V2019[49–51], which was used with the Charmm36[52] forcefield parameters for all subsequent dynamics calculations. Unless otherwise noted, all conditions investigated adhered to the same simulation setup procedures. These steps include solvation of all models using the TIP3P[53,54] water model in rhombic dodecahedron simulation periodic boundaries. Any existing charge was neutralized by addition of $Na^+$ and $Cl^-$ counter ions. Additional ions were added as needed to obtain a 0.1 M NaCl concentration before minimizing each system by steepest descent methods. Convergence of thermodynamic parameters was achieved by a series of simulations up to 5000 picoseconds in length carried out under both NVT and NPT ensembles. Velocity rescaling and Berendsen pressure coupling were used with reference values of 310 K and 1.0 bar, respectively. All subsequent production dynamics were performed in the absence of restraints under an extended NPT ensemble featuring Nose-Hoover[55,56] and Parinhello-Rahman[57,58] temperature and pressure coupling, respectively. All equilibrium simulations were carried out, in triplicate, for 200 ns durations. All MD data have been deposited in the Zenodo public repository.

**Principal component and cross-correlation analyses**. After equilibration by production MD methods, all protein coordinate files were extracted as DCD formatted trajectories for use in the Bio3d V2.3-0[59,60] biological structure analysis R package. Bio3d[59,60] was used for all principal component, proportion of variance, and cross-correlation calculations. Briefly, each structure was simulated according to methods described above and checked for conformational convergence via block averaging of root-mean square deviation (RMSD) measurements along each trajectory. The Visual Molecular Dynamics[61] (VMD) software program V1.9.4a51 was used to produce Bio3d-compatible inputs from the Gromacs simulation data format. New trajectories were then generated by extracting only protein atoms from each frame. Principal components and cross-correlation matrices were then calculated on superposed coordinates of alpha carbons and plotted using R Studio[62].

Principal Component Analysis (PCA) is commonly utilized to reduce multidimensional datasets into simplified systems with lower dimensionality, while also retaining most of the information present in the initial dataset. For MD trajectories, PCA allows the user to identify the principal components that contribute to overall system variance associated with overall groups of correlated atom motions[63–67].

**Center-of-mass (COM) pulling and umbrella sampling**. Estimates of DNA-binding affinities to the repressor complex were assessed by COM pulling and subsequent umbrella sampling of configurations along a reaction coordinate ($\xi$). The DNA-bound model was subjected to MD equilibration for 200 ns to allow for relaxation from constraints associated with crystallization. The DNA-bound model to be investigated was equilibrated by production MD for 200 ns in the Gromacs[49–51] environment using the Charmm36 forcefield as described above. Structurally converged frames were extracted and used as input for COM pulling simulations. Models were solvated with TIP3P[53,54] water in simulation boxes measuring $7.5 \times 7.5 \times 15.0$ nm to allow enough distance along the z-axis to completely dissociate the DNA. Neutralizing counter ions were added, and the system was energy minimized using the steepest descent protocol. Protein and DNA atoms were next restrained during a brief thermodynamic equilibration carried out under an NPT ensemble with a Berendsen barostat. Restraints were then removed from the DNA and COM pulling was accomplished by an applied potential along $\xi$ with a force constant of $1000$ kJ mol$^{-1}$ nm$^2$ at a rate of $0.01$ nm ps$^{-1}$. Umbrella sampling methods were applied using configurations along $\xi$ extracted at COM distances between 0.1 and 0.2 nm. 27 configuration windows were generated in total. Each one was subjected to a short NPT ensemble equilibration and simulated by production molecular dynamics without position restraints for 10 ns. An umbrella potential with a force constant of $1000$ kJ mol$^{-1}$ nm$^2$ was used to ensure adequate sampling of each configuration. Force measurements were collected and used as input for the weighted histogram analysis method (WHAM) in Gromacs[68]. Completeness of sampling was confirmed by visual assessment of umbrella histograms. Estimation of simulated error was determined by bootstrapping methods ($n = 200$).

***Mycobacterium smegmatis* RNA polymerase: repressor complex modeling**. To generate the model of the RNA polymerase:repressor complex, the duplex DNA from the repressor crystal structure was aligned with the duplex DNA portion present in the structure of the *M. smegmatis* transcription initiation complex with a full transcription bubble (PDB 5VI5). Such a model positions the repressor in the duplex DNA region immediately downstream of the transcription bubble to mimic an interaction that would occur with repressor bound at a stoperator site.

**Reporting summary**. Further information on research design is available in the Nature Research Reporting Summary linked to this article.

## Data availability

Coordinates and structure factors for the native and selenomethionine crystal structures have been deposited in the PDB under accession codes 7R6R and 7TZ1, respectively. SAXS data and atomistic models have been deposited in the SASBDB database as entries SASDMK3, SASDML3, and SASDMM3. All MD data have been deposited in the Zenodo public repository (https://doi.org/10.5281/zenodo.6604542). The TipsytheTRex genome sequence has been deposited in GenBank under accession code MF919536. Whole-genome sequencing data have been deposited in the NCBI BioProject database under accession code PRJNA818041. The plaque assays and DNA-binding data generated in this study are provided in the Supplementary Information/Source Data file. The uncropped gels for Supplementary Figs. 4, 6, and 8 are shown in Supplementary Figs. 9, 10, and 11 respectively. Source data are provided with this paper.

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

## Acknowledgements

The authors would like to thank George Meigs and James Holton, beamline 8.3.1 at the Advanced Light Source (ALS), for their assistance in X-ray data collection and processing. Beamline 8.3.1 at the ALS is operated by the University of California Office of the President, Multicampus Research Programs and Initiatives grant MR-15-328599, the National Institutes of Health (R01 GM124149 and P30 GM124169), Plexxikon Inc., and the Integrated Diffraction Analysis Technologies program of the US Department of Energy Office of Biological and Environmental Research. SAXS data was collected at the SIBYLS beamline at the Advanced Light Source (Berkeley, CA) which is supported by the National Institutes of Health (P30 GM124169), and the Integrated Diffraction Analysis Technologies program of the US Department of Energy Office of Biological and Environmental Research. The Advanced Light Source is a national user facility operated by Lawrence Berkeley National Laboratory on behalf of the US Department of Energy under contract DE-AC02-05CH11231, Office of Basic Energy Sciences. This work was supported by the Howard Hughes Medical Institute's Science Education Alliance as well as by funds provided by Western Carolina University (M.D.G. and J.R.W.). Additional support was provided by the Middle Tennessee State University Molecular Biosciences (MOBI) Doctoral program (J.M.M. and C.A.B.). Access to the Voltron Computational Cluster was generously provided by the MTSU Department of Chemistry. We are grateful to WCU students in the 2015–2016 Phage Hunters and 2021 Biochemistry Laboratory courses for their assistance in data collection, Megan Eckardt for assistance with the virus discovery course at WCU, Brittania Bintz for assistance with genome sequencing, and to Graham Hatfull for critical reading of the manuscript.

## Author contributions

R.J.M., B.S., K.N.G., T.J.H., and J.R.W. carried out X-ray crystallization, data collection, structure determination, and refinement. T.C.T. carried out the AlphaFold modeling, density modification, and docking. C.A.B. and J.M.M. performed and analyzed all molecular dynamics data. M.H., R.M., and J.R.W. collected and analyzed SEC–SAXS–MALS data. E.R.C., W.C.G., and M.D.G. designed and carried out the in vivo superinfection immunity assays. J.R.W. performed all DNA-binding experiments. M.D.G. collected and analyzed all

whole-genome sequencing data. M.D.G. and J.R.W. designed and supervised the research. J.M.M., M.D.G., and J.R.W. wrote the manuscript, with input from all authors.

## Competing interests

The authors declare no competing interests.
