## [Peer Review File · Nature Communications]

REVIEWER COMMENTS

Reviewer #1 (Remarks to the Author):

The manuscript by McGinnis et al. presents a crystal structure and biophysical analysis of a dual-function Mycobacterium bacteriophage protein bound to an asymmetric DNA consensus site. In addition to a role as repressor of lytic growth at operator sites within promoter regions, the protein is postulated to terminate transcription at intergenic, directional 'stopoperator' sites. Structure of the monomeric protein is novel in presenting one canonical helix-turn-helix (hth) DNA-binding domain and a second, variant hth domain that binds in the adjacent opening of the major groove on the same face of the DNA. SAXS data support a solution structure for the complex that is consistent with the x-ray data. Molecular dynamics simulations are used to analyse motions within and between domains. The research team is to be credited for producing rigorous and relevant results at institutions known primarily for the education of undergraduate students. Still, although the results are certainly interesting, the overall impact of the work appears to be more suitable for a more specialized journal. In support of that possibility, some detailed but on-balance minor comments are offered here that the authors may find useful.

Several issues should be clarified with respect to the crystallographic results reported in Table 1:

The resolution of 3.13 Å is relatively poor. I encourage a reevaluation of the resolution limit according to Karplus PA, Diederichs K. Science. 2012;336(6084):1030-1033. The native data could perhaps be extended further, although the poor completeness of the high-resolution data could be limiting, but might be avoided by a collection at larger rotation angles if radiation damage is not limiting, all of which should be stated (perhaps in lieu of re-analysis). At least it might be stated whether analysis of higher-resolution data enable more residues to be modeled.

It is not expected that the SeMet data show higher resolution than the native. If there is a reason specific to this system or their methods that offers an explanation for this finding, it should be stated.

A few details in Table 1 should be addressed:

1. line 'Resolution', please add the symbol '*' behind the first bracket to indicate that this is the value for the highest-resolution shell
2. Line 'Wavelength' please review rules for consistent usage of decimal places vs. significant figures, e.g., 0.979690 vs. 1.11608
3. lines 'Total Reflections' and 'Unique Reflections' and below in the refinement statistics the comma as separator is confusing; no mark or just a gap instead would be easier for the reader. Same also with number of atoms below in the refinement stats

4. In data collection statistics: Please provide Wilson B-factor
5. CC1/2 should be CC1/2
6. line 'B-factors' please clarify units and/or average B-factors
7. Lines 'Bond length' and 'Bond angles' entries are statistical values; please indicate which, e.g., r.m.s.

The choices of colors in the figures is sometimes unfortunate. Blue, green, and cyan (Fig 2) are scarcely distinguishable on my screen, and yellow (Fig 3, 7) is almost always a poor choice for visibility, as confirmed here. The panel D legend entry must define the colors, and also the orientation compared with panels a, b, c.

In Fig 2 the use of both r and D in panel B, and R in panel C, end up being confusing; at a minimum all should be defined in the legend. Panel D needs its own legend entry.

In Fig 3 the indication of protein-DNA contacts as lines seems quite bold considering a) the low resolution of the present structure and b) the general fact that bonding contacts involving protons are not directly observable. To support the lines drawn it would be reasonable to include a tabulation of the closest distances and angles between heavy atoms proposed as hydrogen-bond donors/acceptors.

In Fig 4 the relevant patterns would be easier to observe in the aligned sequences if the alignment matched (i.e., aligned with) the sequence logos above.

In Fig 7 the orientation needs to be specified with respect to the views in Figs 1 and 3 and also to the domains of the polymerase.

There are several places in the Results section where speculations are included, and these are generally unwarranted, or at least should be labeled as such, and perhaps included as Discussion instead. For example, first 6 lines on page 9; lines 19-20 on page 13; line 21 on page 16 thru line 5 on page 17; lines 8 -12 on page 17, where it seems there is also a conceptual issue unless stoperator sites have been identified in host genes, which is not documented or referenced.

It is no surprise that structural comparison of the disordered distal domain with other structures turns up no matches. Nevertheless, a sequence comparison may be informative, taking advantage of the structural definition of the domain offered by the authors' structure. In other words, use only the sequence distal to the linker helix as the query, to avoid bias from the N-terminal domain and linker, which may otherwise dominate the search.

Also no surprise is that when two parts of a protein are connected their contributions to binding are non-additive. This is a well-known property even for small molecules, and reflects (at a minimum) the fact that each part need not conduct an independent three-dimensional search for the binding site. This universal effect does not appear to be the kind of synergy the authors imagine.

A couple of technical issues:

Mixing of equimolar amounts of two DNA single strands need not result in complete conversion to duplex, and should be checked (e.g., by monitoring the reaction using hypochromicity). In addition, the authors' annealing conditions are uniquely poor: Tris buffer has one of the highest known thermal coefficients of the pK of any common biological buffer (~0.03 per degree). Thus, depending on the temperature at which the stock pH of Tris is determined, there could be as much as 3 pH units change during annealing, which could alter the protonation state of cytidine residues, and thus their base-pairing potential. Of course they see correct duplexes in the crystals, but these may be a minor component of a complex mixture, a possibility that may limit the success of crystallogensis.

On lines 14-15 on page 21 the authors say that adding 40 mM MgCl₂ eliminated cloudiness in the crystallization drops. Two points: their conclusion that doing so "helped solubilize the complex" is unwarranted, as they have no idea what effect it had other than clearing the cloudiness; and for clarity of documentation they should indicate whether 40 mM was the final concentration of added MgCl₂, which seems very unlikely.

The authors use the phrase 'adjacent DNA major grooves' but this is incorrect: there is only one major groove, and 'adjacent' is in any case wrong. The correct phrasing is 'successive openings of the major groove on the same face of the DNA.'

By MD the authors find that the DNA-bound form of the proteins is less dynamic than the DNA-free form. This too is unsurprising. If there are bona fide examples of increased dynamics of proteins upon ligand binding, those could be discussed and cited here.

The authors refer to other DNA-binding assays; such results would usefully enrich the analysis of binding here, especially with 13-bp DNA, which does cause some change in the SAXS profile.

Reviewer #2 (Remarks to the Author):

As the authors point out, we have only started to scratch the surface on the variety of molecular mechanisms among bacteriophage lysogenic switches, which are of extreme scientific interest both from a fundamental and applied standpoint. The manuscript presents a fascinating new addition.

The manuscript is well written and presents an integrative structural biology approach including crystallography, SAXS and MD simulations, coupled to biological assays, and generally I think it is worthy of publication.

However I think that, to present a complete story, one thing that is missing, is direct measurement of DNA binding affinity (by EMSA, SPR, ITC or other favourite technique) for different variants. This is particularly important to support the the supposed cooperativity of the two domains on DNA binding, which is currently not experimentally tested as far as I can see. Without this, the computational work, which is described at length, seems too speculative. Even for testing the role of different residues, the biological assay used - which is otherwise extremely commendable - gives only an indirect measure, as residue substitutions can also result in misfolding, leading to loss of function. As the structural integrity and stability of the variants cannot be checked in the biological assay, this limits the interpretation of the results. The authors should strongly consider measuring the DNA binding affinity by a chosen method for individual domain constructs and selected point variants, to better support their claims and the computational analysis, or at least explain clearly why this is not feasible.

Another important point is that - given the limited resolution of the crystal structure of the native protein:DNA, it is somewhat difficult to understand the authors did not fully refine and deposit the SeMet structure. The R-factors are quite high, and so are B-factors, and while I am convinced the structure is fundamentally right (protein:DNA complexes are often troublesome) it would be desirable to use all available information. It is stated in the methods: 'The selenomethionine structure was refined to an Rwork/Rfree of 35.1%/39.2%, then used as a model for molecular replacement against the higher-quality native dataset'. However, the statistics in Table 1 suggest that the SeMet version has produced better data (better resolution, overall I/sigma, completeness, and still acceptable other statistics). I would urge the authors to also fully refine and deposit this structure. I understand that it is less biologically relevant, but the better data quality should allow proper refinement and add confidence in the overall study.

Here are some other comments and suggestions that in my opinion would improve the manuscript.

Introduction: perhaps the authors could give an extra few lines of background on other molecular mechanisms of lysogenic switches other than lambda to give a better impression on the variety found in bacteriophages.

Results and discussion

page 7: It would be helpful to number helices in Figure 1 - clearly denoting already here in the the scaffolding and recognition helices referred to later. The numbered helices can then be used

throughout the text, which at times gets longwinded in referring to the helices. Probably it would help the reader to make a short comparison to the lambda CI-NTD already here, rather than later in the text, and perhaps show the lambda CI dimer bound to DNA as comparison in Fig 1.

page 7 line 18: it is a little debatable what constitutes a novel fold. DALI does not perform fantastically with small motifs, and to me, this seems a variation of the HTH motif more than a novel fold. Thus, I would suggest to the authors that they 'tone down' this aspect here and throughout the paper. Their findings are novel and interesting enough, without claiming a novel fold. For example down in page 8 line 17, one can argue that the authors are showing themselves a motif with structural similarity binding DNA in the superposition right in Fig 1C.

SAXS:page 9, line 13 'is consistent' should be changed to 'is most consistent'. Also below, I am somewhat concerned by the discrepancies between experimental and theoretical molecular weights, though I agree on the overall conclusions. Has SDS-PAGE or even better mass spectrometry been run on these samples to show that there is no proteolysis, or are there other good explanations for the deviation? How does the Mw compare to what can be calculated from the SAXS data? I appreciate the advantage of coupling SAXS with SEC-MALS, but a concentration series, or at least report of presumed concentration in the peaks analysed by SAXS would have been useful. Monomer-oligomer state are dependent on concentration, so this would be very informative. What about carrying out Kratky plot, or EOM analysis to extract more information on dynamics?

Protein:DNA Interactions, pages 10-11. It is particularly here that I think figure numbering matched with the figures would help making the text leaner.

Page 13, lines 19-20 'Such stabilization of a closed repressor

20 structure may allow for physiologic regulation of DNA binding activities.' I find this a bit confusing. What is the 'stabilization' it is referred to? No stabilizing interactions are mentioned here. Or is this just a hypothesis of how the repression could be controlled eg by modulators in the cell?

Page 14, line 14.15 and generally on MD simulations. Is PCA analysis really the best way to show this? What about RMSF analysis? How about attempting to correlate the MD to SAXS analysis (an Rg can be obtained from the MD simulations)?

DNA Binding Energy Involves a Synergistic Effect Between Domains, page 14. As already mentioned above, while I think the MD analysis is really interesting, I find that this section is really begging experimental confirmation of affinity for DNA of the individual domains, vs the full length repressor. There are many techniques to do this, and one wonders why it has not been done. I think, in absence of any experimental confirmation, the description and discussion of MD and free energy analysis is rather excessive and far too speculative (including the discussion of possible role of bridge region).

Page 18, lines 1-2 Details of the modelling must be given. May be I missed but I could not localize them.

Page 21, line 12, as a service to the reader, it might be nice to report also the concentration in mg/mL (this is customary in protein crystallization).

As already noted, it is unclear why in the lines below it is stated that the SeMet crystal data is of worse quality (it does not seem so from Table 1), thus I urge the authors to also complete, deposit and comment on the SeMet structure.

Reviewer #3 (Remarks to the Author):

This is an interesting manuscript reporting the structure of a novel phage repressor. Contrary to what is seen with the classical cI -repressors, this novel repressor is a monomer that has two different DNA binding sites. There are, however, some questions related to the functionality of this repressor that should be addressed for a better understanding of the system. Specifically:

1. It is unclear to this reviewer how these prophages are induced. Are they induced after mitomycin C treatment? The authors make an analogy with the classical repressors, and propose that this new type of repressor may be also proteolyzed in order to remove their activity. It would be interesting to see if the two DNA binding domains, when expressed independently, retain their ability to bind to the DNA. These independent domains could be also expressed simultaneously in a recipient cell, to see if they can also block phage infection. These experiments would provide additional support to the authors' hypotheses, and will validate the last part of the paper where most of the conclusions were not addressed experimentally.
2. The aforementioned studies could be completed expressing a form of the repressor that does not have the C-terminal part of the Stoperator domain, suggested to be important to interact with other proteins. It would be expected that a lysogen expressing this mutant repressor would be repressed but uninducible.

3. The experiment in which different version of the repressor are expressed is interesting, especially the results obtained with the wt repressor. Although there is a clear (and expected) interference, how the authors explain the presence of phage plaques when higher concentrations of the phage lysate was used? One would expect that these are scape mutants. However, since there are multiple places where this repressor can bind to, how the evolve phages have emerged? These mutant phages should be sequenced and analysed.

We are extremely grateful to all reviewers for their constructive feedback, as their thoughtful comments and suggestions have helped to improve our manuscript. Below we have offered responses to all reviewer comments, which can be found in bold type

REVIEWER COMMENTS

Reviewer #1 (Remarks to the Author):

The manuscript by McGinnis et al. presents a crystal structure and biophysical analysis of a dual-function Mycobacterium bacteriophage protein bound to an asymmetric DNA consensus site. In addition to a role as repressor of lytic growth at operator sites within promoter regions, the protein is postulated to terminate transcription at intergenic, directional 'stopoperator' sites. Structure of the monomeric protein is novel in presenting one canonical helix-turn-helix (hth) DNA-binding domain and a second, variant hth domain that binds in the adjacent opening of the major groove on the same face of the DNA. SAXS data support a solution structure for the complex that is consistent with the x-ray data. Molecular dynamics simulations are used to analyse motions within and between domains. The research team is to be credited for producing rigorous and relevant results at institutions known primarily for the education of undergraduate students. Still, although the results are certainly interesting, the overall impact of the work appears to be more suitable for a more specialized journal. In support of that possibility, some detailed but on-balance minor comments are offered here that the authors may find useful.

Several issues should be clarified with respect to the crystallographic results reported in Table 1:

The resolution of 3.13 Å is relatively poor. I encourage a reevaluation of the resolution limit according to Karplus PA, Diederichs K. Science. 2012;336(6084):1030-1033. The native data could perhaps be extended further, although the poor completeness of the high-resolution data could be limiting, but might be avoided by a collection at larger rotation angles if radiation damage is not limiting, all of which should be stated (perhaps in lieu of re-analysis). At least it might be stated whether analysis of higher-resolution data enable more residues to be modeled.

It is not expected that the SeMet data show higher resolution than the native. If there is a reason specific to this system or their methods that offers an explanation for this finding, it should be stated.

We thank the reviewer for this comment. Consistent with the recommendations by Karplus and Diederichs, we relied on the CC(1/2) parameter to determine the high-resolution cutoff for both our native and SeMet datasets. Below we show the output from XDS data processing of the native data, with our selected resolution cutoff highlighted in yellow. Based on this analysis, we feel confident that we cannot further extend the resolution of these data and obtain maps that would allow for additional, more accurate modeling of our side chains.

SUBSET OF INTENSITY DATA WITH SIGNAL/NOISE >= -3.0 AS FUNCTION OF RESOLUTION													
RESOLUTION LIMIT	NUMBER OF REFLECTIONS OBSERVED	NUMBER OF REFLECTIONS UNIQUE	NUMBER OF REFLECTIONS POSSIBLE	COMPLETENESS OF DATA	R-FACTOR observed	R-FACTOR expected	COMPARED	I/SIGMA	R-meas	CC(1/2)	Anomal Corr	SigAno	Nano
10.51	3013	267	268	99.6%	4.3%	6.3%	3013	36.98	4.6%	100.0*	6	0.796	183
7.43	5784	453	455	99.6%	6.1%	6.8%	5784	34.69	6.3%	99.9*	-4	0.800	373
6.07	6744	580	580	100.0%	9.6%	8.9%	6744	24.00	10.1%	99.8*	1	0.933	491
5.26	8593	672	676	99.4%	11.0%	10.4%	8593	22.81	11.5%	99.6*	-2	0.864	587
4.70	9897	765	765	100.0%	15.9%	14.8%	9897	17.87	16.5%	99.6*	1	0.902	682
4.29	10146	835	836	99.9%	20.0%	18.6%	10145	14.46	20.9%	99.7*	3	0.906	743
3.97	11642	891	891	100.0%	29.5%	27.8%	11641	12.04	30.8%	98.9*	2	0.877	807
3.72	13212	983	986	99.7%	48.8%	48.3%	13209	8.07	50.7%	98.6*	2	0.793	894
3.50	12855	1038	1039	99.9%	74.7%	75.0%	12852	5.40	78.0%	96.3*	1	0.756	943
3.32	13409	1067	1072	99.5%	101.4%	104.1%	13408	3.97	105.8%	96.9*	4	0.748	974
3.17	14776	1130	1150	98.3%	188.6%	197.1%	14773	2.27	196.6%	91.6*	-1	0.689	1033
3.04	15864	1191	1196	99.6%	468.9%	517.1%	15862	0.97	487.7%	46.6*	-5	0.576	1106
2.92	15453	1213	1227	98.9%	1003.8%	1116.2%	15452	0.36	1046.0%	5.7	0	0.549	1128

2.81	13613	1273	1310	97.2%	2995.3%	3428.6%	13612	0.15	3153.1%	-4.4	-3	0.499	1179
2.71	9086	1210	1335	90.6%	-99.9%	-99.9%	9077	0.00	-99.9%	-9.3	6	0.489	1038
2.63	3339	1013	1403	72.2%	-99.9%	-99.9%	3191	0.00	-99.9%	-20.4	0	0.447	392
2.55	944	588	1396	42.1%	-99.9%	-99.9%	598	0.00	-99.9%	-15.9	-9	0.483	29
2.48	251	242	1458	16.6%	565.2%	323.4%	18	0.22	799.4%	-60.4	0	0.000	0
2.41	122	120	1494	8.0%	515.7%	1024.2%	4	-0.16	729.3%	0.0	0	0.000	0
2.35	47	47	1577	3.0%	-99.9%	-99.9%	0	0.00	-99.9%	0.0	0	0.000	0
total	168790	15578	21114	73.8%	20.8%	22.1%	167873	7.25	21.8%	100.0*	1	0.700	12582

We agree with the reviewer that it is unusual to have a higher resolution SeMet structure as compared to native datasets. The native and SeMet data provided in this manuscript represent the best data we have been able to collect after screening hundreds of crystals on both in-house and synchrotron X-ray sources. We attempted to improve the diffraction quality of these crystals by trying different crystallization conditions, DNA substrates, and different freezing conditions, but we could not improve the resolution above what we have presented in this report.

During manuscript revision, we were fortunate enough to collaborate with Dr. Tom Terwilliger (Los Alamos National Laboratory and Principal Investigator in the Phenix project) to further improve the refinement of both the native and SeMet structures. Dr. Terwilliger reached out to us given the challenging nature of our data, and with his help we were able to combine AlphaFold modeling with structure refinement. Our data served as a test set for the Phenix team to determine how AlphaFold could be used to improve challenging structures, prior to its release in Phenix version 1.20. We have found that including AlphaFold in structure building and refinement significantly improved refinement statistics (see revised Table I), and in the revised manuscript we describe how AlphaFold was used to aid in structure refinement. Given his contributions, Dr. Terwilliger has been added as a co-author to this manuscript.

The DNA binding domains look identical both before and after the new refinement method. Phenix/AlphaFold was able to significantly help us model the distal portion of the stoperator domain that does not bind DNA. In the newly refined structures, we now see a continuous chain for this region, and we see more defined secondary structure in the distal portion of the stoperator domain. We have deposited both the native and SeMet structures to the PDB, and the new Table I has complete data collection and refinement statistics for both structures.

A few details in Table 1 should be addressed:

1. line 'Resolution', please add the symbol '*' behind the first bracket to indicate that this is the value for the highest-resolution shell
2. Line 'Wavelength' please review rules for consistent usage of decimal places vs. significant figures, e.g., 0.979690 vs. 1.11608
3. lines 'Total Reflections' and 'Unique Reflections' and below in the refinement statistics the comma as separator is confusing; no mark or just a gap instead would be easier for the reader. Same also with number of atoms below in the refinement stats
4. In data collection statistics: Please provide Wilson B-factor
5. CC1/2 should be CC1/2
6. line 'B-factors' please clarify units and/or average B-factors
7. Lines 'Bond length' and 'Bond angles' entries are statistical values; please indicate which, e.g., r.m.s.

We have updated Table I to both include the new refinement statistics and to address the above comments. We have a few specific comments:

1. For point 2., the wavelength values we have reported are as defined by the beamline during data collection.

2. For point 5., we are not sure what change the reviewer is recommending regarding the CC1/2 parameter.

The choices of colors in the figures is sometimes unfortunate. Blue, green, and cyan (Fig 2) are scarcely distinguishable on my screen, and yellow (Fig 3, 7) is almost always a poor choice for visibility, as confirmed here. The panel D legend entry must define the colors, and also the orientation compared with panels a, b, c.

To improve the visibility of the figures, we have generated a new figure 2 that uses a different coloring pattern that is easier to view. For figure 3, we have removed the yellow bond between D104 and R108 and replaced this with a more visible red bond.

To address the additional concern related to figure 7 (below), we changed the color scheme of the RNA polymerase such that rather than being all gray, panel 7A shows the RNA polymerase now colored by protein subunit. We have also removed the yellow coloring.

In Fig 2 the use of both r and D in panel B, and R in panel C, end up being confusing; at a minimum all should be defined in the legend. Panel D needs its own legend entry.

We have included additional language in the figure legend to clarify these parameters, and to provide a legend for panel D.

In Fig 3 the indication of protein-DNA contacts as lines seems quite bold considering a) the low resolution of the present structure and b) the general fact that bonding contacts involving protons are not directly observable. To support the lines drawn it would be reasonable to include a tabulation of the closest distances and angles between heavy atoms proposed as hydrogen-bond donors/acceptors.

To further clarify this figure, and to provide a reasonable representation of interactions between the DNA and protein given our resolution, we have set a cutoff distance of 3.2 Angstroms for all polar contacts between protein and DNA. We analyzed both the native and SeMet structures for protein:DNA contacts, and residues that do not have an asterisks are contacts that are observed in both structures. Those that have an asterisk are interactions only observed in the higher resolution structure (we did not see any contacts in the lower-resolution native structure that were not conserved in the SeMet structure). Therefore, all residues in figure 3A that are providing polar contacts are less than 3.2 Angstroms and are within hydrogen bonding distance of the DNA substrate. We have adjusted the figure legend to clarify the parameters for these interactions.

In Fig 4 the relevant patterns would be easier to observe in the aligned sequences if the alignment matched (i.e., aligned with) the sequence logos above.

We thank the reviewer for this suggestion. We have moved this particular portion of the figure to supplemental (please see Figure S6), and have adjusted the figure such that the consensus sequences observed in the genome align with the sequence logo.

In Fig 7 the orientation needs to be specified with respect to the views in Figs 1 and 3 and also to the domains of the polymerase.

To clarify the different domains of the polymerase, we changed the color scheme such that rather than being all gray, panel 7A shows the RNA polymerase now colored by protein subunit.

In the current figure 7 we provide an arrow that shows the direction of transcription, and this same arrow is present in both figures 1 and 3. Additionally, in figure 7, the repressor is color coded the same as in figures 1 and 3. The orientation of the repressor has been rotated by ~90 degrees (relative to figure 1) in order to clearly show the position of the beta' subunit relative to the stoperator domain. To clarify this further, we have included text in the manuscript to describe the rotation of figure 7 relative to figure 1.

There are several places in the Results section where speculations are included, and these are generally unwarranted, or at least should be labeled as such, and perhaps included as Discussion instead. For example, first 6 lines on page 9; lines 19-20 on page 13; line 21 on page 16 thru line 5 on page 17; lines 8-12 on page 17, where it seems there is also a conceptual issue unless stoperator sites have been identified in host genes, which is not documented or referenced.

In our manuscript we combined the results and discussion into one cohesive section, so the points raised here by the reviewer are indeed speculations on our part. Our intent with these statements is to attempt to correlate our results with the bigger picture of repressor function. For page 17, lines 8-12, we are specifically discussing how the repressor halts transcription initiation or elongation of viral genes by the host RNA polymerase. We do not believe that the repressor also silences host transcription, and indeed there is no data to support that conclusion. We have clarified this in the text to make it clear we focused on models for the silencing of viral genes.

It is no surprise that structural comparison of the disordered distal domain with other structures turns up no matches. Nevertheless, a sequence comparison may be informative, taking advantage of the structural definition of the domain offered by the authors' structure. In other words, use only the sequence distal to the linker helix as the query, to avoid bias from the N-terminal domain and linker, which may otherwise dominate the search.

We thank the reviewer for this comment. Along with the DALI analysis, we also performed detail BLAST analyses of the stoperator domain, including the full stoperator domain, the DNA binding portion of the stoperator, and the distal C-terminal portion of the stoperator domain that we predict may bind partner proteins. The results were quite intriguing, as we found many viral and bacterial sequences that contain this motif (it is unclear if the bacterial sequences are indeed bacterial or are from prophages). The majority of sequences are annotated as either an immunity repressor or something generic (hypothetical protein) and are ~180 residues in length. What was quite interesting is that we also found sequences where the full repressor protein is fused to other protein domains, including a helicase and a Serpin (serine proteinase inhibitor) motif. Therefore, along with a functional role as an immunity repressor, this protein appears to have evolved to work alongside other functions. We also observed viral sequences that contained just the stoperator domain as its own polypeptide, and in many cases these viral genomes also contain the N-terminal HTH portion as a separate gene. It would be quite interesting to know if these proteins form a complex once expressed, although our new DNA binding studies reveal that the individual HTH and stoperator domains in our repressor fail to form a complex and bind DNA when expressed separately (see comments below).

In all of the analyses, we failed to find another protein containing this domain that has been biochemically or structurally characterized. Therefore, our results provide the first description of this domain.

Also no surprise is that when two parts of a protein are connected their contributions to binding are non-additive. This is a well-known property even for small molecules, and reflects (at a minimum) the fact

that each part need not conduct an independent three-dimensional search for the binding site. This universal effect does not appear to be the kind of synergy the authors imagine.

We agree that the reviewer is likely correct that this observation is the consequence of one domain influencing binding of the other via removal of the need for both domains to independently search in three-dimensions for two separate binding sites. We do not expect that our observations reflect classical allostery in the sense that DNA binding at one site induces a conformational change at a secondary site. With this in mind, we have edited the manuscript language to clarify this point. Lastly, we highlight newly introduced experimental data that clearly show cooperative behavior for repressor binding to DNA. We have also included new binding data that demonstrates the isolated domains fail to bind DNA. This result contrasts our MD data wherein the isolated complexes are computationally forced to begin the simulation as an intact complex. For this reason, in the revised text we have removed the free energy calculations for the isolated domains.

A couple of technical issues:

Mixing of equimolar amounts of two DNA single strands need not result in complete conversion to duplex, and should be checked (e.g., by monitoring the reaction using hypochromicity). In addition, the authors' annealing conditions are uniquely poor: Tris buffer has one of the highest known thermal coefficients of the pK of any common biological buffer (~0.03 per degree). Thus, depending on the temperature at which the stock pH of Tris is determined, there could be as much as 3 pH units change during annealing, which could alter the protonation state of cytidine residues, and thus their base-pairing potential. Of course they see correct duplexes in the crystals, but these may be a minor component of a complex mixture, a possibility that may limit the success of crystallogensis.

In this study we used annealed DNA substrates for 1) the crystal structures, 2) SAXS studies, and 3) DNA binding studies (which were recommended by the reviewers and have been included in the revised manuscript). All duplex DNAs were annealed using the same buffer described in the methods. To confirm that the annealing reaction worked as anticipated, we monitored the annealing of a fluorescently-labeled ssDNA substrate to its complement (the same optimal DNA used for our DNA binding studies), and compared its migration to that of a labeled ssDNA in a 5% native gel. The gel presented below shows that our annealing reaction did indeed work, as we see a loss of the ssDNA band and the formation of duplex DNA that migrates higher in the native gel as compared to ssDNA.

Given that we see clear electron density for the duplex DNA in our crystal structure, and that our EMSA work monitoring DNA binding shows bands consistent with duplex DNA formation, we feel confident that the results presented in this manuscript all represent interactions between the repressor and duplex DNA.

On lines 14-15 on page 21 the authors say that adding 40 mM MgCl₂ eliminated cloudiness in the crystallization drops. Two points: their conclusion that doing so "helped solubilize the complex" is unwarranted, as they have no idea what effect it had other than clearing the cloudiness; and for clarity of documentation they should indicate whether 40 mM was the final concentration of added MgCl₂, which seems very unlikely.

During crystallization trials, we observed that the clear protein solution turned cloudy when mixed with the DNA substrate at room temperature, and the solution remained cloudy after a 30 minute incubation. The addition of 40 mM MgCl₂ caused the solution to clear, and as a result our initial crystal drops were clear. We have modified the text to state that the addition of 40 mM MgCl₂ eliminated the observed cloudiness. 40 mM was indeed the concentration of MgCl₂ added.

The authors use the phrase 'adjacent DNA major grooves' but this is incorrect: there is only one major groove, and 'adjacent' is in any case wrong. The correct phrasing is 'successive openings of the major groove on the same face of the DNA.'

We thank the reviewer for pointing this out and have included a revised statement in the manuscript.

By MD the authors find that the DNA-bound form of the proteins is less dynamic than the DNA-free form. This too is unsurprising. If there are bona fide examples of increased dynamics of proteins upon ligand binding, those could be discussed and cited here.

We are unaware of related examples where the bound-state is more dynamic than the apo state. However, the MD work has been included here to support both our crystallization (no crystals were observed in the absence of DNA) and SAXS studies in that DNA binding reduces the conformational flexibility of the repressor.

The authors refer to other DNA-binding assays; such results would usefully enrich the analysis of binding here, especially with 13-bp DNA, which does cause some change in the SAXS profile.

We thank the reviewer for this comment, as it greatly enhances our manuscript. We have performed DNA binding studies of the repressor protein on three different DNA substrates: a 30-bp "optimal" substrate that contains the 13-mer consensus sequence, a "scrambled" substrate where the 30-bp optimal sequence was scrambled to remove the consensus, and a 13-bp substrate that contains only the consensus sequence. The results presented in the revised manuscript show that the wild-type repressor binds the 30-bp optimal but not the scrambled sequence, which agrees with previous reports on the L5 repressor. Additionally, we see that the 13-bp substrate is too small for

the repressor to efficiently bind. This result is in strong agreement with our SAXS data where we could not observe a stable complex on the 13-bp DNA, as well as our crystal structure that shows some residues (W50 for example) bind the DNA outside of the consensus sequence.

Using the optimal DNA substrate, we also looked at the DNA binding affinities of the mutant repressor proteins that showed a strong phenotype in the in vivo superinfection immunity assays (Figure 4). These results are presented in the revised text and agree with the in vivo results. All mutants tested show a loss of DNA binding with the exception of D104A.

We also looked at the DNA binding affinities of the HTH and stoperator domains when separately expressed, and our results show that the individual HTH and stoperator domains do not bind DNA. We also observe a loss of DNA binding when the separate HTH and stoperator domains are mixed, which indicates that the domains must be fused together in a single polypeptide in order to efficiently bind the consensus sequence.

Reviewer #2 (Remarks to the Author):

As the authors point out, we have only started to scratch the surface on the variety of molecular mechanisms among bacteriophage lysogenic switches, which are of extreme scientific interest both from a fundamental and applied standpoint. The manuscript presents a fascinating new addition.

The manuscript is well written and presents an integrative structural biology approach including crystallography, SAXS and MD simulations, coupled to biological assays, and generally I think it is worthy of publication.

However I think that, to present a complete story, one thing that is missing, is direct measurement of DNA binding affinity (by EMSA, SPR, ITC or other favourite technique) for different variants. This is particularly important to support the the supposed cooperativity of the two domains on DNA binding, which is currently not experimentally tested as far as I can see. Without this, the computational work, which is described at length, seems too speculative. Even for testing the role of different residues, the biological assay used - which is otherwise extremely commendable - gives only an indirect measure, as residue substitutions can also result in misfolding, leading to loss of function. As the structural integrity and stability of the variants cannot be checked in the biological assay, this limits the interpretation of the results. The authors should strongly consider measuring the DNA binding affinity by a chosen method for individual domain constructs and selected point variants, to better support their claims and the computational analysis, or at least explain clearly why this is not feasible.

We thank the reviewer for this suggestion, as we agree that DNA binding studies will greatly enhance the impact of our manuscript. To address this concern, we have performed a variety of EMSA DNA binding assays, and these results have been included in the revised manuscript. The highlights of the results are as follows:

-We show that the wild-type repressor binds only a DNA sequence that contains the consensus motif. It does not recognize a duplex DNA that lacks the consensus. Additionally, we show that just the 13-bp consensus motif is not sufficient for binding by the repressor; the repressor needs more than just the consensus in order to bind (see new Figure S4). This result agrees well with our SAXS data (figure 2), where we do not observe a stable complex on a 13-bp DNA containing the consensus motif, and with our crystal structure (Figure 3), where we observe residues (W50, for example) that bind the DNA outside of the consensus.

-We tested DNA binding of mutant repressor proteins that show a significant phenotype in the superinfection immunity assays presented in figure 4 (see new Figure 4 and Figure S5). All of these point mutants were soluble when expressed in *E. coli* and behaved like the wild-type protein during purification, and all mutants tested showed a loss of DNA binding with the exception of D104A. The D104A mutant shows weaker DNA binding as compared to wild-type, but it can still form a specific complex on DNA. Our DNA binding results also revealed that the wild-type and D104A repressor proteins bind DNA cooperatively. We discuss all of these results in the revised manuscript.

We also tested whether the HTH and Stoperator domains, when expressed separately, can bind DNA. Our results show that the individual HTH and Stoperator domains do not bind DNA (see new Figure S7). Additionally, we also see no DNA binding when the HTH and Stoperator domains are mixed, which provides evidence that the two domains must be fused into a single polypeptide for stable DNA binding.

Another important point is that - given the limited resolution of the crystal structure of the native protein:DNA, it is somewhat difficult to understand the authors did not fully refine and deposit the SeMet structure. The R-factors are quite high, and so are B-factors, and while I am convinced the structure is fundamentally right (protein:DNA complexes are often troublesome) it would be desirable to use all available information. It is stated in the methods: 'The selenomethionine structure was refined to an Rwork/Rfree of 35.1%/39.2%, then used as a model for molecular replacement against the higher-quality native dataset'. However, the statistics in Table 1 suggest that the SeMet version has produced better data (better resolution, overall I/sigma, completeness, and still acceptable other statistics). I would urge the authors to also fully refine and deposit this structure. I understand that it is less biologically relevant, but the better data quality should allow proper refinement and add confidence in the overall study.

We fully agree with the reviewer's statement here, and we have now refined and deposited both the native and SeMet structures. Working with this data has proven challenging; however, during manuscript revision we were fortunate enough to collaborate with Dr. Tom Terwilliger (Los Alamos National Laboratory and Principal Investigator in the Phenix project) to further improve both the native and SeMet structures. Dr. Terwilliger reached out to us given the challenging nature of our data, and with his help we were able to combine AlphaFold modeling with structure refinement. Our data served as a test set for the phenix team to observe how AlphaFold could be used to improve challenging structures, prior to its release in Phenix version 1.20. We have found that including AlphaFold in the structure building and refinement significantly improved refinement statistics (see revised Table I), and in the revised manuscript we describe how AlphaFold was used to aid in structure refinement. Given his contributions, Dr. Terwilliger has been added as a co-author to this manuscript.

The DNA binding domains look identical both before and after AlphaFold. AlphaFold was able to significantly help us model the distal portion of the stoperator domain that does not bind DNA. In the newly refined structures, we now see a continuous chain for this region, and we see more defined secondary structure for the distal portion of the stoperator domain. Both the fully refined native and SeMet structures were analyzed during manuscript revision, especially in regards to residues involved in DNA binding.

Here are some other comments and suggestions that in my opinion would improve the manuscript.

Introduction: perhaps the authors could give an extra few lines of background on other molecular mechanisms of lysogenic switches other than lambda to give a better impression on the variety found in bacteriophages.

In the revised text we have added additional information regarding different mechanisms of bacteriophage lysogenic switches.

Results and discussion

page 7: It would be helpful to number helices in Figure 1 - clearly denoting already here in the the scaffolding and recognition helices referred to later. The numbered helices can then be used throughout the text, which at times gets longwinded in referring to the helices. Probably it would help the reader to make a short comparison to the lambda CI-NTD already here, rather than later in the text, and perhaps show the lambda CI dimer bound to DNA as comparison in Fig 1.

Our revised figure 1 has all secondary structural elements labeled, and this labeling matches the sequence alignment presented in supplemental Figure S1. Additionally, we have included a structure of the CI dimer bound to DNA as a new panel C in Figure 1, which allows the reader to compare the CI dimer to our structure presented in panels A and B. We have modified the text to refer to the new labeling when discussing various helices.

page 7 line 18: it is a little debatable what constitutes a novel fold. DALI does not perform fantastically with small motifs, and to me, this seems a variation of the HTH motif more than a novel fold. Thus, I would suggest to the authors that they 'tone down' this aspect here and throughout the paper. Their findings are novel and interesting enough, without claiming a novel fold. For example down in page 8 line 17, one can argue that the authors are showing themselves a motif with structural similarity binding DNA in the superposition right in Fig 1C.

In the revised text we have reworded to emphasize that rather than this being a novel fold, the DNA binding portion of the stoperator domain has a fold that is a variation of the HTH motif.

SAXS:page 9, line 13 'is consistent' should be changed to 'is most consistent'. Also below, I am somewhat concerned by the discrepancies between experimental and theoretical molecular weights, though I agree on the overall conclusions. Has SDS-PAGE or even better mass spectrometry been run on these samples to show that there is no proteolysis, or are there other good explanations for the deviation? How does the Mw compare to what can be calculated from the SAXS data?

It is difficult to accurately determine molecular weight by MALS for a small 23kDa protein:DNA complex that has a partially unfolded region(s). Thus, the molecular weights from our MALS/SAXS data should be considered an estimate of the oligomeric state. The SAXS-based molecular weight for the protein:DNA complex (new Supplementary Table S1) is a significant under-estimate, as the DNA scatters three-time stronger than the protein. Despite these difficulties, the molecular weights from both MALS and SAXS, together with a good match of our atomistic models to the experimental SAXS data (Figure 2C), clearly support that the repressor binds DNA as a monomer.

SDS-PAGE was run on this protein preparation, and we did not see evidence of significant proteolysis. Additionally, the long tail in the P(r) function (Figure 2B) and normalized Kratky plot (new Supplementary Figure S2) validate the presence of unfolded region(s) in the repressor protein. Therefore, we do not believe that a proteolytically cleaved repressor accounts for the lower molecular weights seen by MALS/SAXS. We have added additional text to the manuscript to clarify this result.

I appreciate the advantage of coupling SAXS with SEC-MALS, but a concentration series, or at least report of presumed concentration in the peaks analysed by SAXS would have been useful. Monomer-oligomer state are dependent on concentration, so this would be very informative.

It is unusual to collect SEC-SAXS-MALS data with various sample concentrations. In our experiment, we injected 50uL of ~5.8 mg/ml (~ 270uM, see Methods) sample with minimal concentration to obtain a good signal for an ~23kDa protein. The 1:5 dilution upon SEC leads to an ~1.2 mg/ml sample at SAXS data collection. This low concentration led to an interference-free SAXS profile, as judged by the Guinier plot (Figure 2C), and was free of nonspecific oligomerization or structure factors often exhibited at higher protein concentrations.

What about carrying out Kratky plot, or EOM analysis to extract more information on dynamics?

We thank the reviewer for this suggestion and have added a normalized Kratky plot in the new supplemental Figure S2. This data, along with the long tail in the P(r) function, support the presence of the unfolded region(s) in the repressor protein.

Our modeling conformational sampling using BILBOMD and model selection using MultiFOXS explored the conformational flexibility of the unfolded N-terminal ~40 residues (see Methods). This large unfolded region significantly contributes to the SAXS scattering profile and prevented us from further extracting more dynamic information on the core of the protein.

Protein:DNA Interactions, pages 10-11. It is particularly here that I think figure numbering matched with the figures would help making the text leaner.

We have used the new labeling in Figure 1 to reduce the redundancy for this region of the text.

Page 13, lines 19-20 'Such stabilization of a closed repressor structure may allow for physiologic regulation of DNA binding activities.' I find this a bit confusing. What is the 'stabilization' it is referred to? No stabilizing interactions are mentioned here. Or is this just a hypothesis of how the repression could be controlled eg by modulators in the cell?

Our intent here was to state that in the absence of DNA the HTH and Stoperator domains can interact in order to generate a closed form of the protein that cannot bind DNA, and this may serve as a mechanism to regulate the DNA binding activity of the protein. In the revised text we have reworded this statement to say that the adoption of a closed repressor may serve to regulate DNA binding activities.

Page 14, line 14.15 and generally on MD simularions. Is PCA analysis really the best way to show this? What about RMSF analysis? How about attempting to correlate the MD to SAXS analysis (an R_g can be obtained from the MD simulations)?

Correlating the SAXS data with our MD results is unfortunately not possible. The MD simulations were performed without the unfolded N-region present, which hinders comparison of R_g estimates derived from each method. The SAXS modeling were performed with the protein containing this region. Furthermore, the SAXS and all-atom MD experiments were performed on significantly different timescales, where all-atom MD simulations are restricted to ns-ms timescales. Therefore, the MD simulations are not always capable of recapitulating experimental results; a common example can be found in the field of protein folding. In order to better correlate these two methods, additional simulations would be needed for comparison.

Regarding PCA analysis, PCA captures the variance associated with each atom in the system such that global conformational motions can be identified and modeled to represent motions that are statistically likely. In contrast, RMSF analysis represents the motion of individual atoms or residues averaged over the course of the entire simulation and provides no information on actual conformational dynamics. RMSF is appropriate to capture local movements, but not entire domain motions.

DNA Binding Energy Involves a Synergistic Effect Between Domains, page 14. As already mentioned above, while I think the MD analysis is really interesting, I find that this section is really begging experimental confirmation of affinity for DNA of the individual domains, vs the full length repressor. There are many techniques to do this, and one wonders why it has not been done. I think, in absence of any experimental confirmation, the description and discussion of MD and free energy analysis is rather excessive and far too speculative (including the discussion of possible role of bridge region).

We have revised our MD analysis in order to discuss the data in light of our new DNA binding results. The most substantial change relates to the newly introduced experimental data that clearly shows that the isolated HTH and Stoperator domains fail to bind DNA. This result contrasts our MD data wherein the isolated complexes are computationally forced to begin the simulation as an intact complex. For this reason, in the revised text we have removed the free energy calculations for the isolated domains.

Our DNA binding data show cooperative behavior for full-length repressor binding to DNA, and in the revised text we use our MD data to offer an explanation for this behavior. The results of Figure 6 show that that HTH and Stoperator domains have different DNA binding energies, with the Stoperator domain dissociating from the DNA first. The two-fold difference in binding energies calculated for each DNA binding domain would promote step-wise binding events that would lead to sigmoidal DNA binding behavior. Therefore, our MD data allow us to propose a model in which the HTH domain could bind the DNA first, thus allowing the Stoperator domain to engage its sequence without having to do a full independent three-dimensional search.

Page 21, line 12, as a service to the reader, it might be nice to report also the concentration in mg/mL (this is customary in protein crystallization).

We have added this information to the Methods section.

As already noted, it is unclear why in the lines below it is stated that the SeMet crystal data is of worse quality (it does not seem so from Table 1), thus I urge the authors to also complete, deposit and comment on the SeMet structure.

We have now fully refined and deposited the SeMet crystal structure. While we were able to solve the structure by SAD using this data, in our initial refinement of this structure our Rfactors were hung up in the mid to upper 30%. Two things significantly helped the SeMet structure: First, when we reanalyzed the SeMet data we noticed that some of the X-ray beam was peeking out from the beam stop, and this caused high Rfactors in our lowest resolution bin. We reprocessed the data using HKL3000 and removed this portion of the low-resolution data. Second, thanks to the new contributions of Dr. Terwilliger, the combination of AlphaFold with refinement allowed us to improve both the SeMet and native structures, and the SeMet structure now has refinement statistics more acceptable for the resolution limit (see Table I).

The native and SeMet data provided in this manuscript represent the best datasets we were able to collect after screening hundreds of crystals on both in-house and synchrotron X-ray sources. We attempted to improve the diffraction quality of these crystals by trying different crystallization conditions, DNA substrates, and different freezing conditions, but we could not improve the resolution above what we have presented in this report. However, we are still confident in both structures, and both were used in the analysis of the protein fold as well as the interactions with the DNA substrate.

Reviewer #3 (Remarks to the Author):

This is an interesting manuscript reporting the structure of a novel phage repressor. Contrary to what is seen with the classical cI-repressors, this novel repressor is a monomer that has two different DNA binding sites. There are, however, some questions related to the functionality of this repressor that should be addressed for a better understanding of the system. Specifically:

1. It is unclear to this reviewer how these prophages are induced. Are they induced after mitomycin C treatment? The authors make an analogy with the classical repressors, and propose that this new type of repressor may be also proteolyzed in order to remove their activity. It would be interesting to see if the two DNA binding domains, when expressed independently, retain their ability to bind to the DNA. These independent domains could be also expressed simultaneously in a recipient cell, to see if they can also block phage infection. These experiments would provide additional support to the authors' hypotheses, and will validate the last part of the paper where most of the conclusions were not addressed experimentally.

Currently, very little is known regarding mycobacteriophage prophage induction. In a 2012 review (Hatfull, G.F. The secret lives of mycobacteriophages. *Adv Virus Res* 82, 179-288 (2012)), Graham Hatfull noted that for mycobacteriophage L5, no other protein has been identified (other than the repressor) that would aid in the lytic-lysogenic decision. The same is true for TispytheTRex (the phage that is the focus of this study), although our new whole-genome sequencing results may have pointed to a gene that aids in lysogeny (discussed below). The review also notes that L5 lysogens are not strongly inducible by DNA damaging agents, which points to these phages using a mechanism that is distinct from most temperate phages. Along with L5 lysogens, that Hatfull laboratory also tried to induce lysogens of several cluster A bacteriophages with a range of mitomycin C concentrations, but had little to no success. In these experiments they saw some phage release, but nothing significant (Graham Hatfull, personal communication). We have added text to the revised manuscript to discuss the lack of knowledge regarding mycobacteriophage prophage induction.

We thank the reviewer for the recommended DNA binding experiments. Based on the suggestion of multiple reviewers to include DNA binding data to support our structure, we have performed a variety of EMSA DNA binding assays, and these results have been included in the revised manuscript. The highlights of the results as related to this comment are as follows:

-We show that the wild-type repressor binds only a DNA sequence that contains the consensus motif. It does not recognize a duplex DNA that lacks the consensus. Additionally, we show that just the 13-bp consensus motif is not sufficient for efficient binding by the repressor; the repressor needs more than just the consensus in order to bind (see new Figure S4). This result agrees well with our SAXS data (Figure 2), where we do not observe a stable complex on a 13-bp DNA containing the consensus motif, and with our crystal structure (Figure 3), where we observe residues (W50, for example) that bind the DNA outside of the consensus.

-We tested whether the HTH and Stoperator domains, when expressed separately, can bind DNA. Our results show that the individual HTH and Stoperator domains do not bind DNA. Additionally, we also see no DNA binding when the HTH and Stoperator domains are mixed (see new Figure S7), which provides evidence that the two domains must be fused into a single polypeptide for stable DNA binding and function. This provides support for our hypothesis that a cleavage event between the HTH and Stoperator domains could be a regulatory mechanism to control repressor function.

Our new data clearly show that the individual HTH and Stoperator domains cannot bind DNA, either alone or when mixed together, and we therefore hypothesize that the individual domains would not block phage infection in vivo. For future work, we are interested in further exploring whether a cleavage event that would separate the two domains is indeed a mechanism of repressor regulation. This of course would be the focus of an additional manuscript.

2. The aforementioned studies could be completed expressing a form of the repressor that does not have the C-terminal part of the Stoperator domain, suggested to be important to interact with other proteins. It would be expected that a lysogen expressing this mutant repressor would be repressed but uninducible.

We attempted to address this comment by expressing a truncated repressor protein that is missing the C-terminal portion of the stoperator domain predicted to bind partner proteins. Our selection of where to truncate (at residue N119) was based on our crystal structure. This residue lies in a loop region that connects alpha helix 6 (part of DNA binding domain) to beta strand 1 (first secondary structural element of the C-terminal region, please see the new Figure 1 that now has all secondary structure labeled), and such a truncation would keep all DNA binding portions of the protein intact while removing the full C-terminal region of the stoperator domain. Subsequent experiments revealed that this truncated protein is not soluble (please see the figure below, comparing the expression and purification of the wild-type repressor (wt in the figure) and the Delta119 truncated mutant (labeled “trunc” in the figure)). The wild-type and truncated proteins were purified side by side on the same day, but were analyzed on different SDS-PAGE gels. As shown in the figure, while the truncated protein is expressed (boxed band in the Induced lane), it is not retained in the soluble fraction (Trunc soluble).

An inspection of the crystal structure reveals many hydrophobic contacts between the DNA binding portion of the stoperator domain (at helices 5 and 6) and the C-terminal portion of the Stoperator domain that is predicted to bind partner proteins, which likely explains the insoluble behavior of our truncated protein. Therefore, it appears that this portion of the protein is required for proper

folding, and we were not able to further explore the function of a repressor protein lacking the C-terminal portion of the stoperator domain.

3. The experiment in which different version of the repressor are expressed is interesting, especially the results obtained with the wt repressor. Although there is a clear (and expected) interference, how the authors explain the presence of phage plaques when higher concentrations of the phage lysate was used? One would expect that these are scape mutants. However, since there are multiple places where this repressor can bind to, how the evolve phages have emerged? These mutant phages should be sequenced and analysed.

To address this, a full plaque assay using the TipsytheTRex stock was performed in WT repressor-expressing cells. Five plaques were picked, and stocks were grown after another round of purification. These viruses were indeed escape mutants, as all five had efficiencies of plating close to 1 when they were spotted onto WT repressor-expressing cells. The genomes of all five repressor escape mutant viruses (REMs) and the WT stock were sequenced and their genomes *de novo* assembled (please see the new Figure S3). The results revealed that complete resistance was conferred by only a single point mutation in each REM. One mutation caused the introduction of a stop codon early in the repressor gene, while the other four caused nonpolar to polar substitutions in the last or 4th to last amino acid of a protein of unknown function produced from gene 74, which is adjacent to the repressor gene (gene 75). These results are in agreement with previous analyses of spontaneous clear plaque mutants and assorted cluster A lysogen and repressor escape mutants (Donnelly-Wu, M.K., Jacobs, W.R., Jr. & Hatfull, G.F. Superinfection immunity of mycobacteriophage L5: applications for genetic transformation of mycobacteria. *Mol Microbiol* 7, 407-17 (1993), and Mavrich, T.N. & Hatfull, G.F. Evolution of Superinfection Immunity in Cluster A Mycobacteriophages. *mBio* 10(2019)). In these published mutants and in our study, no evidence of mutation of the operator/stoperator sites occurred, but instead resistance was usually conferred by deletions or mutations in the area of the repressor gene itself. This is in contrast to the lambda system and likely reflects the large number of operator/stoperator sites that would need to be mutated simultaneously to observe a loss of sensitivity to the WT repressor.

REVIEWER COMMENTS

Reviewer #1 (Remarks to the Author):

The authors have addressed most prior comments adequately; I cannot recall why I noted CC1/2 but the Table seems fine now.

However there is now a major problem with the data and interpretation of Figure 4. It is patently impossible for a 1:1 monomer:DNA interaction to display cooperative behavior with respect to monomer concentration, as Fig 4C seems to show and is interpreted to show. However the x axis scale is very odd, making it impossible to understand if that is true, or if not, what may be wrong. Fig 4C appears to resemble a semi-logarithmic plot, but it is impossible to tell if it is because of the choice of axis scale. ALL binding data are sigmoidal on semi-logarithmic plots, including binding with NO cooperativity at all. The authors mention in their reply to reviews Hill coefficients of ~ 2 - ~ 3 (referencing Fig 4 but none are shown there). Hill coeffs. have many well known faults, and there is no need for them. Only the raw data should be analysed, which Fig 4C attempts to do, but its x axis scale makes analysis impossible. A linear x axis should be used, and also separately a semilogarithmic one, with the same data plotted on both plots, and both require sensible x-axis increments.

Simply put, as long as the two domains are attached to each other and bind the DNA as a monomer, they CANNOT display any higher-order concentration dependence except by some artifact. They may well synergize each other's binding, but this effect CANNOT be manifested as an apparently cooperative concentration dependence. The fact that it seems to do so indicates that something is very wrong; in my opinion the two most likely possibilities are that there is no cooperativity when analysed correctly, or more than one repressor monomer binds to each DNA in a stoichiometric titration to repressor excess.

Although the stoichiometric titration data of Fig 4D appear to be convincing alone, the x axis suggests that repressor was titrated into DNA, but the legend says saturation occurs at 1:1 ratio of DNA to protein. Obviously if cooperativity with respect to monomer concentration is really present then the protein:DNA ratio at saturation must be greater than 1:1, but DNA:protein at saturation will still be 1:1. Ideally both directions of the titration should be carried out.

Regrettably, the issues raised are so severe that I cannot approve the ms until they are rectified.

Reviewer #2 (Remarks to the Author):

The authors have put considerable experimental, analysis and rewriting efforts in improving the weak parts of their original manuscript. In particular binding to DNA has been experimentally verified for a number of constructs, and the crystal structures quality has been considerably improved. I have no further comments and congratulate the authors on their work, which I consider very suitable for publication in Nature Communications.

Reviewer #3 (Remarks to the Author):

This is a very nice piece of work. The authors have brilliantly addressed all my previous comments. One last minor question: since some of the evolved phages (insensitive to the expression of the repressor) have mutations in gene 74, one would anticipate that these escape mutants cannot produce lysogens. Has this been tested experimentally?

As I said, congratulations for such beautiful work.

We are extremely grateful to all reviewers for their constructive feedback, as their thoughtful comments and suggestions have helped to improve our manuscript. Below we have offered responses to all reviewer comments, which can be found in bold type

REVIEWER COMMENTS

Reviewer #1 (Remarks to the Author):

The authors have addressed most prior comments adequately; I cannot recall why I noted CC1/2 but the Table seems fine now.

However there is now a major problem with the data and interpretation of Figure 4. It is patently impossible for a 1:1 monomer:DNA interaction to display cooperative behavior with respect to monomer concentration, as Fig 4C seems to show and is interpreted to show.

We thank the reviewer for pointing out this error in data interpretation. After reviewing the literature and discussing the data further, we have refined the manuscript to remove any conclusion of these data stating cooperativity. The best explanation, based on general thermodynamics and our literature review, would suggest that the sigmoidal shape of our reported binding isotherms is the result of a conformational distribution between low- and high-affinity apo states. Conceptually, this is similar to the MWC model commonly applied to hemoglobin. In the edited manuscript, we propose that the apo protein resides in a conformational equilibrium as described and summarized in the figure below. Here, we highlight that DNA binding preferentially to the apo R_o state would require a shift in the equilibrium population towards promotion of additional R_o species. Such conformational equilibria have been proposed previously to explain similar sigmoidal behavior for monomeric enzymes (Porter and Miller. "Cooperativity in Monomeric Enzymes with Single Ligand-Binding Sites. *Bioorg. Chem.* 2012.).

The manuscript has been updated to remove general references to cooperativity and to introduce the following paragraph:

“Taken together, these data provide a possible explanation for the observation of sigmoidal character in DNA binding isotherms (Figure 4C, Inset). Not typically observed for monomeric proteins²⁸, sigmoidal shape often results from cooperative ligand binding. However, the plot of fraction bound versus [repressor]:[DNA] (Figure 4D) clearly indicates a 1:1 binding stoichiometry, which is not consistent with cooperative binding for a monomeric protein. Instead, this sigmoidal shape is likely the consequence of a conformational equilibrium between low- and high-affinity states that populates independent of DNA. Indeed, our MD analysis indicates that the apo protein adopts distinct “closed” and “open” conformations that limit access to the DNA binding site (Figure 5C-5E). Preferential DNA binding to the high-affinity “open” conformation would promote an equilibrium shift away from the low-affinity binding state as required by Le Chatelier’s principle and consequently yield sigmoidal binding isotherms.”

However the x axis scale is very odd, making it impossible to understand if that is true, or if not, what may be wrong. Fig 4C appears to resemble a semi-logarithmic plot, but it is impossible to tell if it is because of the choice of axis scale. ALL binding data are sigmoidal on semi-logarithmic plots, including binding with NO cooperativity at all. The authors mention in their reply to reviews Hill coefficients of ~2 - ~3 (referencing Fig 4 but none are shown there). Hill coefs. have many well known faults, and there is no need for them. Only the raw data should be analysed, which Fig 4C attempts to do, but its x axis scale makes analysis impossible. A linear x axis should be used, and also separately a semilogarithmic one, with the same data plotted on both plots, and both require sensible x-axis increments.

Figure 4 has been updated to present the experimental binding data using a linear x-axis. We have included an inset to highlight the sigmoidal character of the data observed at lower repressor concentration. To place emphasis on the sigmoidal shape of the binding curve, we have included the Hill coefficients since the parameter does prove useful to diagnosing sigmoidal behaviors. While we agree that the parameter can itself be problematic with respect to interpretation, it is nonetheless useful to diagnose whether a dataset is hyperbolic or truly sigmoidal. Figure 4 is included below.

Simply put, as long as the two domains are attached to each other and bind the DNA as a monomer, they CANNOT display any higher-order concentration dependence except by some artifact. They may well synergize each other's binding, but this effect CANNOT be manifested as an apparently cooperative concentration dependence. The fact that it seems to do so indicates that something is very wrong; in my opinion the two most likely possibilities are that there is no cooperativity when analysed correctly, or more than one repressor monomer binds to each DNA in a stoichiometric titration to repressor excess.

After consideration of the reviewer's comments, we agree that the sigmoidal shape of the binding data is likely not a consequence of cooperativity or any higher-order concentration dependence as described. Given the observed 1:1 binding stoichiometry and conformational distribution noted from Molecular Dynamics simulations, we favor an explanation that posits the sigmoidal shape is due to a conformational switch from low- to high-affinity states. This does not require any conclusion of cooperativity. Therefore, we favor the reviewer's first conclusion that DNA binding occurs non-cooperatively.

Although the stoichiometric titration data of Fig 4D appear to be convincing alone, the x axis suggests that repressor was titrated into DNA, but the legend says saturation occurs at 1:1 ratio of DNA to protein. Obviously if cooperativity with respect to monomer concentration is really present then the protein:DNA ratio at saturation must be greater than 1:1, but DNA:protein at saturation will still be 1:1. Ideally both directions of the titration should be carried out.

Given the discussion points above, we agree that this result further supports that this is not due to cooperativity, and we have removed such language in the revised manuscript. Our intent with Figure 4D is to simply offer further support for both our structure and SAXS data, which both support a monomer of repressor binding the DNA.

We have not performed a reverse titration (vary DNA against a fixed concentration of repressor) due to issues of experimental design. The binding assay we report is gel-based and relies on fluorescein emissions derived from labeled DNA ligands. This assay is our only option due to currently available resources at our institution. All DNA ligands in-hand are obtained commercially with FAM conjugation already performed. We have opted to vary the repressor concentration and keep the DNA concentration fixed in order to maintain a constant signal in our gel shift assay. If FAM-labeled DNA were varied, binding would be obscured due to changing overall loading mass of DNA. The reverse experiment would thus require a titration of unlabeled DNA against fluorescently-labeled repressor, and we currently do not have a reliable protocol in hand for efficient, single labeling of the protein.

Future work will be needed to experimentally discern the underlying cause of the reported sigmoidal character. However, we also acknowledge a noticeable lack of information in the literature on the topic due to difficulty in measuring such conformational changes.

Regrettably, the issues raised are so severe that I cannot approve the ms until they are rectified.

We hope that we have sufficiently addressed all of the reviewer's concerns to warrant publication in *Nature Communications*.

Reviewer #2 (Remarks to the Author):

The authors have put considerable experimental, analysis and rewriting efforts in improving the weak parts of their

original manuscript. In particular binding to DNA has been experimentally verified for a number of constructs, and the crystal structures quality has been considerably improved. I have no further comments and congratulate the authors on their work, which I consider very suitable for publication in Nature Communications.

We thank the reviewer for their constructive feedback during the initial revision, as they greatly improved our manuscript.

Reviewer #3 (Remarks to the Author):

This is a very nice piece of work. The authors have brilliantly addressed all my previous comments. One last minor question: since some of the evolved phages (insensitive to the expression of the repressor) have mutations in gene 74, one would anticipate that these escape mutants cannot produce lysogens. Has this been tested experimentally? As I said, congratulations for such beautiful work.

We thank the reviewer for their comments. Their recommendation for looking for REMs has greatly improved our manuscript and provided a possible avenue for understanding the regulation of lysogeny via gene 74. We predict that the REM's will not be able to form lysogens, however this as well as the biochemical characterization of gp74 will be the focus of future work.

REVIEWER COMMENTS

Reviewer #1 (Remarks to the Author):

The authors have addressed most prior comments adequately; I cannot recall why I noted $CC_{1/2}$ but the Table seems fine now. However there is now a major problem with the data and interpretation of Figure 4. It is patently impossible for a 1:1 monomer:DNA interaction to display cooperative behavior with respect to monomer concentration, as Fig 4C seems to show and is interpreted to show.

We are extremely grateful to all reviewers for their constructive feedback, as their thoughtful comments and suggestions have helped to improve our manuscript. Below we have offered responses to all reviewer comments, which can be found in bold type

REVIEWER COMMENTS

Reviewer #1 (Remarks to the Author):

The authors have addressed most prior comments adequately; I cannot recall why I noted CC1/2 but the Table seems fine now.

However there is now a major problem with the data and interpretation of Figure 4. It is patently impossible for a 1:1 monomer:DNA interaction to display cooperative behavior with respect to monomer concentration, as Fig 4C seems to show and is interpreted to show.

We thank the reviewer for pointing out this error in data interpretation. After reviewing the literature and discussing the data further, we have refined the manuscript to remove any conclusion of these data stating cooperativity. The **best** explanation, based on general thermodynamics and our literature review, would suggest that the sigmoidal shape of our reported binding isotherms is the result of a conformational distribution between low- and high-affinity apo states. Conceptually, this

The only way such a shift could explain the data is if its time scale is significantly shorter than that of binding, resulting in two populations of protein that do not interchange with each other on the time scale of the binding process. Most, but not all, conformational transitions, being intramolecular, occur in times far shorter than those requiring three-dimensional diffusion, as binding does. To sustain their interpretation would require some analysis of kinetics, which goes beyond the scope of this ms. Instead the authors should tone down their explanation as being not 'best' as here or 'likely' as below, but they must also remove the incorrect comments about Le Chatelier's principle, and measure the disappearance of free DNA, as detailed below.

is similar to the MWC model commonly applied to hemoglobin. In the edited manuscript, we propose that the apo protein resides in a conformational equilibrium as described and summarized in the figure below. Here, we highlight that DNA binding preferentially to the apo R_o state would require a shift in the equilibrium population towards promotion of additional R_o species. Such conformational equilibria have been proposed previously to explain similar sigmoidal behavior for monomeric enzymes (Porter and Miller. "Cooperativity in Monomeric Enzymes with Single Ligand-Binding Sites. *Bioorg. Chem.* 2012.).

The manuscript has been updated to remove general references to cooperativity and to introduce the following paragraph:

“Taken together, these data provide a possible explanation for the observation of sigmoidal character in DNA binding isotherms (Figure 4C, Inset). Not typically observed for monomeric proteins²⁸, sigmoidal shape often results from cooperative ligand binding. However, the plot of fraction bound versus [repressor]:[DNA] (Figure 4D) clearly indicates a 1:1 binding stoichiometry, which is not consistent with cooperative binding for a monomeric protein. Instead, this sigmoidal shape is **likely** the consequence of a conformational equilibrium between low- and high-affinity states that populates independent of DNA. Indeed, our MD analysis indicates that the apo protein adopts distinct “closed” and “open” conformations that limit access to the DNA binding site (Figure 5C-5E). Preferential DNA binding to the high-affinity “open” conformation would promote an equilibrium shift away from the low-affinity binding state as required by Le Chatelier’s principle and

Conformational adjustments, being intramolecular as noted above, do not trigger a change in mass according to Le Chatelier. Rather, the binding equilibrium simply 'waits' for more of the binding-competent state to form. This effect is manifested kinetically, not at equilibrium, unless the conformational process is extremely slow. One assumes that if it were slow enough to create discrete populations of the protein, then it would not be observable on the MD time scale either.

consequently yield sigmoidal binding isotherms.”

However the x axis scale is very odd, making it impossible to understand if that is true, or if not, what may be wrong. Fig 4C appears to resemble a semi-logarithmic plot, but it is impossible to tell if it is because of the choice of axis scale. ALL binding data are sigmoidal on semi-logarithmic plots, including binding with NO cooperativity at all. The authors mention in their reply to reviews Hill coefficients of ~2 - ~3 (referencing Fig 4 but none are shown there). Hill coefs. have many well known faults, and there is no need for them. Only the raw data should be analysed, which Fig 4C attempts to do, but its x axis scale makes analysis impossible. A linear x axis should be used, and also separately a semilogarithmic one, with the same data plotted on both plots, and both require sensible x-axis increments.

Figure 4 has been updated to present the experimental binding data using a linear x-axis. We have included an inset to highlight the sigmoidal character of the data observed at lower repressor concentration. To place emphasis on the sigmoidal shape of the binding curve, we have included the Hill coefficients since the parameter does prove useful to diagnosing sigmoidal behaviors. While we agree that the parameter can itself be problematic with respect to interpretation, it is nonetheless useful to diagnose whether a dataset is hyperbolic or truly sigmoidal. Figure 4 is included below.

Simply put, as long as the two domains are attached to each other and bind the DNA as a monomer, they CANNOT display any higher-order concentration dependence except by some artifact. They may well synergize each other's binding, but this effect CANNOT be manifested as an apparently cooperative concentration dependence. The fact that it seems to do so indicates that something is very wrong; in my opinion the two most likely possibilities are that there is no cooperativity when analysed correctly, or more than one repressor monomer binds to each DNA in a stoichiometric titration to repressor excess.

After consideration of the reviewer's comments, we agree that the sigmoidal shape of the binding data is likely not a consequence of cooperativity or any higher-order concentration dependence as described. Given the observed 1:1 binding stoichiometry and conformational distribution noted from Molecular Dynamics simulations, we favor an explanation that posits the sigmoidal shape is due to a conformational switch from low- to high-affinity states. This does not require any conclusion of cooperativity. Therefore, we favor the reviewer's first conclusion that DNA binding occurs non-cooperatively.

The data of Figures 4B and 4D suggest another possible interpretation that seems to me far more likely. The y-axis of Figure 4D suggests that quantitation of the bound fraction relies on the optical densities of the upper bound forms shown in panel B. By eye, D104A does not bind significantly before lane 4. It is very difficult to make reliable determinations of such low optical densities. The result may well differ if instead the disappearance of free DNA were measured, and this should certainly be done for comparison.

Although the stoichiometric titration data of Fig 4D appear to be convincing alone, the x axis suggests that repressor was titrated into DNA, but the legend says saturation occurs at 1:1 ratio of DNA to protein. Obviously if cooperativity with respect to monomer concentration is really present then the protein:DNA ratio at saturation must be greater than 1:1, but DNA:protein at saturation will still be 1:1. Ideally both directions of the titration should be carried out.

Given the discussion points above, we agree that this result further supports that this is not due to cooperativity, and we have removed such language in the revised manuscript. Our intent with Figure 4D is to simply offer further support for both our structure and SAXS data, which both support a monomer of repressor binding the DNA.

We have not performed a reverse titration (vary DNA against a fixed concentration of repressor) due to issues of experimental design. The binding assay we report is gel-based and relies on fluorescein emissions derived from labeled DNA ligands. This assay is our only option due to currently available resources at our institution. All DNA ligands in-hand are obtained commercially with FAM conjugation already performed. We have opted to vary the repressor concentration and keep the DNA concentration fixed in order to maintain a constant signal in our gel shift assay. If FAM-labeled DNA were varied, binding would be obscured due to changing overall loading mass of DNA. The reverse experiment would thus require a titration of unlabeled DNA against fluorescently-labeled repressor, and we currently do not have a reliable protocol in hand for efficient, single labeling of the protein.

Now that the authors have clarified that the stoichiometric titration they report was indeed done to protein excess and yields a 1:1 result, there is no real need for the reverse titration. However I would point out for their future reference that the reverse titration would not demand labeled protein, nor would it necessitate excesses of labeled DNA that would confound the signal. The experiment is conducted using a trace amount of labeled DNA together with increasing amounts of unlabeled DNA to increment the total DNA concentration.

Future work will be needed to experimentally discern the underlying cause of the reported sigmoidal character. However, we also acknowledge a noticeable lack of information in the

literature on the topic due to difficulty in measuring such conformational changes.

Another possible reason for scarcity of literature is that conformational change does not explain results like theirs, as I have argued above. I would still maintain that the most likely source for the sigmoidal behavior lies in some kind of error of experimental design or interpretation, both of which are addressed above. As such I feel they should greatly tone down their explanation. If they will insist to leave it in then it becomes more reasonable to insist on some kinetic evidence to support it. It would be great if they would be the first to document such an effect.

Regrettably, the issues raised are so severe that I cannot approve the ms until they are rectified.

We hope that we have sufficiently addressed all of the reviewer's concerns to warrant publication in *Nature Communications*.

Reviewer #2 (Remarks to the Author):

The authors have put considerable experimental, analysis and rewriting efforts in improving the weak parts of their

original manuscript. In particular binding to DNA has been experimentally verified for a number of constructs, and the crystal structures quality has been considerably improved. I have no further comments and congratulate the authors on their work, which I consider very suitable for publication in Nature Communications.

We thank the reviewer for their constructive feedback during the initial revision, as they greatly improved our manuscript.

Reviewer #3 (Remarks to the Author):

This is a very nice piece of work. The authors have brilliantly addressed all my previous comments. One last minor question: since some of the evolved phages (insensitive to the expression of the repressor) have mutations in gene 74, one would anticipate that these escape mutants cannot produce lysogens. Has this been tested experimentally? As I said, congratulations for such beautiful work.

We thank the reviewer for their comments. Their recommendation for looking for REMs has greatly improved our manuscript and provided a possible avenue for understanding the regulation of lysogeny via gene 74. We predict that the REM's will not be able to form lysogens, however this as well as the biochemical characterization of gp74 will be the focus of future work.